# Adipocyte lipolysis protects mice against *Trypanosoma brucei* infection

Henrique Machado [1], Peter Hofer[2], Rudolf Zechner[2,3,4], Terry K. Smith[5] & Luísa M. Figueiredo [1] ✉

*Trypanosoma brucei* causes African trypanosomiasis, colonizing adipose tissue and inducing weight loss. Here we investigated the molecular mechanisms responsible for adipose mass loss and its impact on disease pathology. We found that lipolysis is activated early in infection. Mice lacking B and T lymphocytes fail to upregulate adipocyte lipolysis, resulting in higher fat mass retention. Genetic ablation of the rate-limiting adipose triglyceride lipase specifically from adipocytes (*Adipoq^{Cre/+}-Atgl^{fl/fl}*) prevented the stimulation of adipocyte lipolysis during infection, reducing fat mass loss. Surprisingly, these mice succumbed earlier and presented a higher parasite burden in the gonadal adipose tissue, indicating that host lipolysis limits parasite growth. Consistently, free fatty acids comparable with those of adipose interstitial fluid induced loss of parasite viability. Adipocyte lipolysis emerges as a mechanism controlling local parasite burden and affecting the loss of fat mass in African trypanosomiasis.

*Trypanosoma brucei* is a unicellular and extracellular parasite that causes human and animal African trypanosomiasis[1]. Within the mammalian host, *T. brucei* colonizes both intravascular and extravascular spaces, causing a progressive wasting disease that is lethal if left untreated[2].

The clinical signs of African trypanosomiasis are associated with the organs where parasites accumulate. This is illustrated by the neurological symptoms observed in humans upon colonization of the central nervous system[3] and the association of skin lesions with the presence of *T. brucei* in the dermis[4].

The loss of fat/adipose mass, which occurs during *T. brucei* infections in the context of a broader wasting syndrome[3], may be associated with the parasite presence in the adipose tissue (AT). This is supported by the disproportionately high accumulation of *T. brucei* in the AT of infected mice[5,6]. Moreover, in the gonadal AT (that is, the largest visceral depot), this leads to a progressive accumulation of myeloid cells as well as tumour necrosis factor α (TNF-α) and interferon-γ-producing lymphocytes[7]. Concomitant to the mounting of this immune response, AT depots undergo a progressive loss of weight[5,7], suggesting that AT colonization by *T. brucei* and the ensuing immune response may be responsible for the loss of fat mass observed in African trypanosomiasis.

Loss of fat mass occurs when lipid catabolism is favoured over lipid anabolism (that is, lipogenesis), resulting in increased degradation of triacylglycerol (TAG) species contained within the adipocyte's lipid droplet[8]. In adipocytes, this mobilization occurs primarily through neutral lipolysis, where TAG is sequentially hydrolysed into diacylglycerol and monoacylglycerol by three lipases: adipose triglyceride lipase (ATGL), hormone-sensitive lipase and monoacylglycerol lipase[8]. The resulting free fatty acids (FFAs) and glycerol molecules are exported to the AT's interstitial spaces and subsequently into the circulatory system.

Multiple stimuli regulate adipocyte lipolysis, including inflammatory molecules (for example, TNF-α)[9], sympathetic nervous system cues (for example, norepinephrine)[10], hormones (for example, catecholamines or insulin)[11] and bacterial products (for example, lipopolysaccharide and peptidoglycans)[12,13]. Proper regulation of adipocyte lipolysis is required for whole-organism energy homoeostasis, especially during times of negative energy balance (for example, fasting)[14]. Importantly, deregulation of adipocyte lipolysis has been shown to

[1]Instituto de Medicina Molecular-João Lobo Antunes, Faculdade de Medicina, Universidade de Lisboa, Lisbon, Portugal. [2]Institute of Molecular Biosciences, University of Graz, Graz, Austria. [3]Field of Excellence BioHealth, University of Graz, Graz, Austria. [4]BioTechMed-Graz, Graz, Austria. [5]School of Biology, Biomedical Sciences Research Complex, University of St Andrews, St Andrews, UK. ✉e-mail: lmf@medicina.ulisboa.pt

promote the development of cancer-associated cachexia[15], insulin resistance[16] and hepatic steatosis[17].

In this Article, we show that loss of fat mass during infection occurs due to an increase in adipocyte lipolysis, particularly through ATGL activity. Moreover, we show that the immune response is an important driver of adipocyte lipolysis and that, in its absence, infected mice are more resistant to fat loss. Using adipocyte-specific ATGL-deficient mice, we established that adipocyte lipolysis has a host-protective function. ATGL-dependent lipolysis prolongs host survival and promotes interstitial FFA accumulation, which reduces local parasite burden (Extended Data Fig. 1).

## Results

### T. brucei infection induces lipolysis in gonadal AT

Natural and experimental infections with *T. brucei* parasites induce weight loss, which includes loss of fat mass. To study this process, the gonadal AT of C57BL/6J mice infected with pleomorphic *T. brucei* was collected at different timepoints post-infection, weighed and subjected to histological analyses. Infected mice showed progressive loss of gonadal AT weight, amounting to a 30%, 54% and 65% loss by days 10, 16 and 30 post-infection, respectively (Fig. 1a). This was consistent with a reduction in individual adipocyte lipid droplet size, which was observed concomitant with a previously described accumulation of inflammatory infiltrates[7] (Fig. 1b,c). Quantification of lipid droplet area revealed a significant average reduction of 44%, 54% and 65%, by days 10, 16 and 30 post-infection, respectively (Fig. 1c).

To test whether loss of fat mass was a result of increased adipocyte lipolysis, gonadal AT from infected and control mice was collected at different timepoints post-infection and incubated ex vivo to allow for the release of lipolytic products, that is, FFAs and glycerol. These were quantified and normalized to the total amount of protein in the tissue. A significant increase in FFA release was observed at days 6 and 9 post-infection (Fig. 1d), indicating that adipocyte lipolysis is stimulated early in infection. Conversely, FFA release was significantly decreased by day 16 post-infection. This later reduction in adipocyte lipolysis stems partially from a normalization bias that occurs at later timepoints of infection, when the AT becomes comparatively more protein rich as lipid stores are depleted and immune infiltration progresses (Extended Data Fig. 2).

Likewise, an increase in glycerol release was observed by days 6 and 9 post-infection relative to non-infected controls, while no differences were observed by day 16 post-infection (Fig. 1e). This difference observed between FFAs and glycerol release by day 16 post-infection may in part be due to glycerol secreted by the parasite, as this metabolite is produced by *T. brucei* during glycolysis, especially under hypoxic conditions[18,19]. Consistently, when *T. brucei* is grown for 24 h in culture, the concentration of FFAs in the medium does not change. In contrast, medium glycerol content increased proportionally to parasite density and was significantly higher under a low oxygen concentration compatible with physiological range found in the AT[20] (Extended Data Fig. 3).

Interestingly, stimulation of AT explants from mice 9 and 16 days post-infection with forskolin (that is, an inducer of lipolysis) failed to elicit the release of lipolytic products to levels comparable with those of non-infected controls (Extended Data Fig. 4a,b). These results suggest that a progressive *T. brucei*-induced AT atrophy may impair adipocyte function.

Overall, these data show that white AT undergoes a progressive weight loss, and adipocyte lipolysis is activated early during *T. brucei* infection, consuming most of its lipid stores by day 16.

### Infection-induced lipolysis requires immune cues

Next, we sought to identify the upstream signals responsible for activation of adipocyte lipolysis. In other contexts, adipocyte lipolysis can be induced by distinct triggers, including immune, neuronal and hormonal signals[8].

To test whether sympathetic innervation contributed to *T. brucei*-induced fat loss, we infected mice that were chemically sympathectomized with 6-hydroxydopamine (6-OHDA), a neurotoxic analogue of dopamine that damages nerve terminals of sympathetic neurons, preventing norepinephrine release[21]. No significant differences were observed in fat mass loss between 6-OHDA-treated and sham-treated infected mice, that is, mice that were injected with only phosphate-buffered saline (PBS) with 0.4% ascorbic acid (Extended Data Fig. 5a), suggesting that changes in sympathetic tone are not involved in loss of fat mass during *T. brucei* infection.

Second, we tested whether infected animals were losing fat mass because of their feeding behaviour. Indeed, we have previously described a transient disease-induced hypophagia around the first peak of parasitaemia (that is, days 4 to 10 post-infection)[5]. To test if hypophagia is sufficient to induce adipocyte lipolysis, we employed two distinct experimental setups of controlled food intake: fasting and re-feeding setup or paired-feeding. In the fasting and re-feeding setup, mice were fasted for 7 h during peak activity hours, followed by a 2-h re-feeding period before being euthanized. Under these conditions, infected mice showed the same early dynamic increase in adipocyte lipolysis, with a reduction after days 10–12 post-infection (Fig. 2a and Extended Data Fig. 6a). In the paired-feeding setup, the food intake of the non-infected mice was limited to that of their respective infected counterparts in a pair-wise scheme (Fig. 2b and Extended Data Fig. 6b). In these conditions, infected mice, but not non-infected pair-fed mice, showed higher adipocyte lipolysis than non-infected ad libitum-fed controls (Fig. 2c and Extended Data Fig. 6c,d), suggesting that disease-induced hypophagia is not sufficient to induce adipocyte lipolysis during *T. brucei* infection.

Next we questioned whether immune activation was required for lipolysis upregulation. To test this hypothesis, we infected recombination activation gene-2 deficient (*Rag2*[−/−]) mice, which lack both B and T cells. These immunosuppressed mice present hyperparasitaemia (Fig. 2d) and usually succumb to infection by day 15 post-infection[22].

Longitudinal body composition analysis of infected *Rag2*[−/−] mice revealed that the loss of fat mass was lower than in infected wild-type (WT) controls (Fig. 2e). Indeed, by day 6 and 9 post-infection, while WT mice presented a 25% and 48% reduction in total fat mass, respectively, *Rag2*[−/−] mice showed no substantial loss of fat mass. By day 12 post-infection, *Rag2*[−/−] mice had lost 14% of total fat mass, significantly less than the 37% loss of fat mass exhibited by infected WT mice at the same time. This reduction in fat mass loss in *Rag2*[−/−] mice was consistent with a reduced capacity to upregulate adipocyte lipolysis (Fig. 2f). While FFA release in infected WT mice increased by 3.25-fold, relative to non-infected mice, this increased only by 1.67-fold in *Rag2*[−/−] mice (Fig. 2f). Intriguingly, AT of infected *Rag2*[−/−] showed a similar release of glycerol to AT of infected WT mice (Extended Data Fig. 6d). This probably stems from an increase in parasite-secreted glycerol due to increased parasite load in the animal, including in the AT[7]. Interestingly, mice deficient for TNF-α, a cytokine classically associated with *T. brucei*-induced wasting and a known activator of adipocyte lipolysis, showed only a small delay in fat mass retention (Extended Data Fig. 5b), suggesting that additional inflammatory factors are involved in fat mass loss and lipolysis induction.

Together, these data show that activation of adipocyte lipolysis and loss of fat mass during *T. brucei* infection is primarily dependent on the presence of T and/or B lymphocytes.

### ATGL activity drives loss of fat mass and adipocyte volume

After establishing that adipocyte lipolysis is increased during *T. brucei* infection, we questioned whether we could revert the fat loss phenotype by deleting the initiating enzyme of neutral lipolysis. For this, we used adipocyte-specific ATGL-deficient mice (*Adipoq*[Cre/+]-*Atgl*[fl/fl])[23,24].

Infected *Adipoq*[Cre/+]-*Atgl*[fl/fl] (knockout (KO)) mice showed a significantly lower secretion of FFAs and glycerol than infected *Atgl*[fl/fl] (WT)

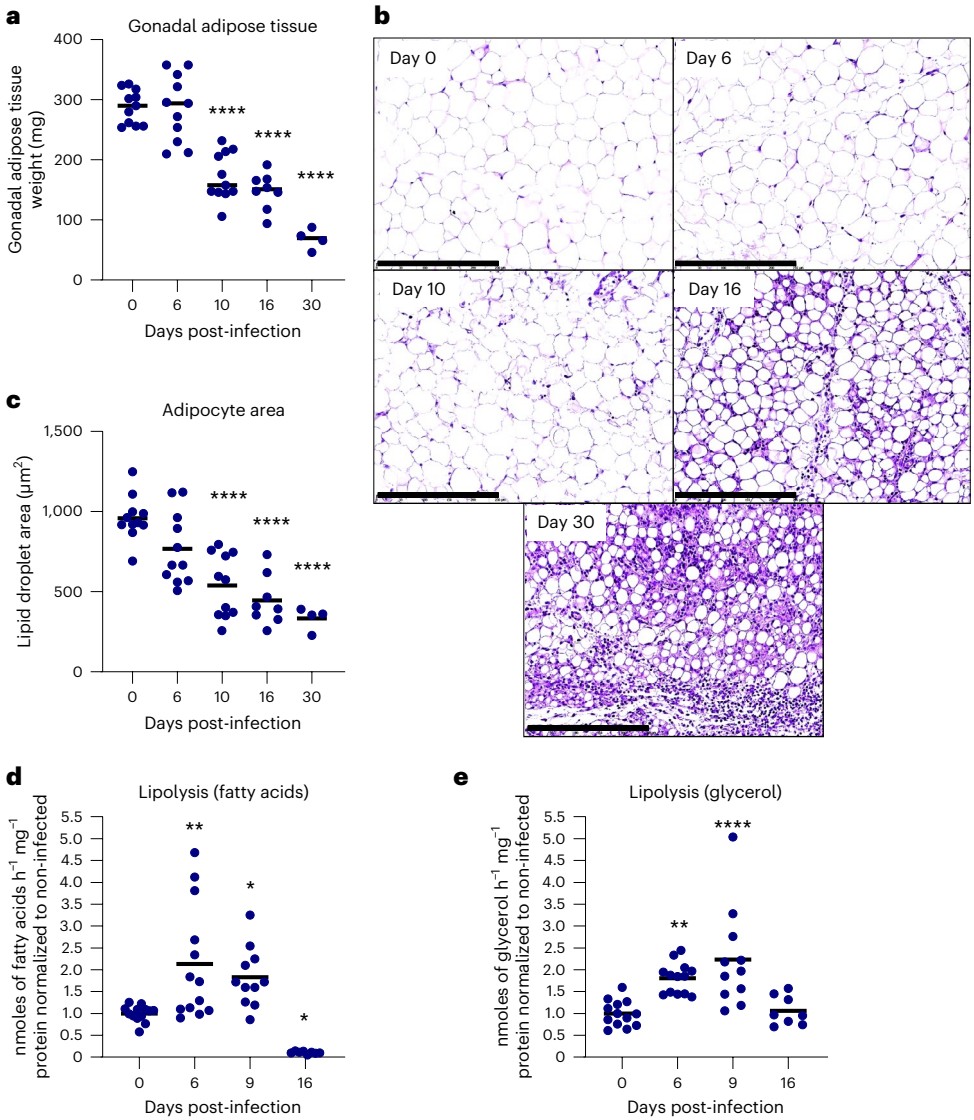

**Fig. 1 | Adipocyte lipolysis is increased during *T. brucei* infection. a**, Weight of gonadal AT throughout infection. **b**, Representative haematoxylin and eosin micrographs of gonadal AT at different timepoints post-infection. Scale bar, 250 μm; magnification, 20×. **c**, Quantification of lipid droplet area of the gonadal AT at different timepoints post-infection. In **a** and **c**, for days 0, 6, 10, 16 and 30, *n* = 11, 11, 11, 8 and 4 mice examined over two independent experiments, respectively. **d,e**, Ex vivo release of FFAs (**d**) and glycerol (**e**) from AT explants collected from mice at different times post-infection relative to non-infected controls. In **d** and **e**, for days 0, 6, 9 and 16, *n* = 13, 13, 11 and 8 mice examined over two independent experiments, respectively. Statistical analysis was performed with one-way ANOVA using Šidák's test for multiple comparisons. *$P < 0.05$, **$P < 0.01$, ****$P < 0.0001$. Statistical source data contain additional parameters.

littermate mice (Fig. 3a and Extended Data Fig. 7a), indicating that ATGL is essential for the increased lipolysis rates observed during *T. brucei* infection.

Consistently, histological analysis of gonadal AT sections revealed that *Adipoq^{Cre/+}-Atgl^{fl/fl}* mice maintained a stable average adipocyte size, that is, ~900 μm², while *Atgl^{fl/fl}* mice experienced a progressive reduction in adipocyte size of up to 75% by day 30 post-infection, showing that adipocyte size reduction is predominantly due to ATGL-dependent lipolysis (Fig. 3b,c). This increased adipocyte size in *Adipoq^{Cre/+}-Atgl^{fl/fl}* mice was consistent with higher gonadal AT depot weights in this group than in *Atgl^{fl/fl}* controls (Fig. 3d). Overall, these data show that increased ATGL-mediated lipolysis in adipocytes is the main mechanism of fat mass loss during *T. brucei* infection.

Next, we explored the effects of ATGL deficiency on mouse body composition. Our hypothesis was that, without ATGL activity, infected mice could not access stored energy in adipocyte TAG. This could trigger alternative catabolic pathways like muscle proteolysis. To

investigate this, we conducted a longitudinal body composition study using nuclear magnetic resonance, tracking body weight, lean, fluid and fat masses.

Total body weight and lean mass of *Atgl^{fl/fl}* and *Adipoq^{Cre/+}-Atgl^{fl/fl}* followed a similar profile, with an initial drop after the first peak of parasitaemia, that is, days 5–8 post-infection, followed by a rapid recovery (Fig. 3e,f), which probably stems from hepatomegaly and splenomegaly observed during *T. brucei* infections[25]. Afterwards, total body weight and lean mass declined for both WT and KO mice, with a trend for *Adipoq^{Cre/+}-Atgl^{fl/fl}* to show lower total weight and lean mass during the latest stages of infection (Fig. 3e,f). During infection, a clear increase in free fluid mass was observed in both WT and KO mice (Fig. 3g), which was probably due to the formation of widespread oedema that stems from the increased vascular leakage known to occur during *T. brucei* infection[6].

Lastly, although both groups experienced loss of fat mass, *Adipoq^{Cre/+}-Atgl^{fl/fl}* mice presented significantly increased retention

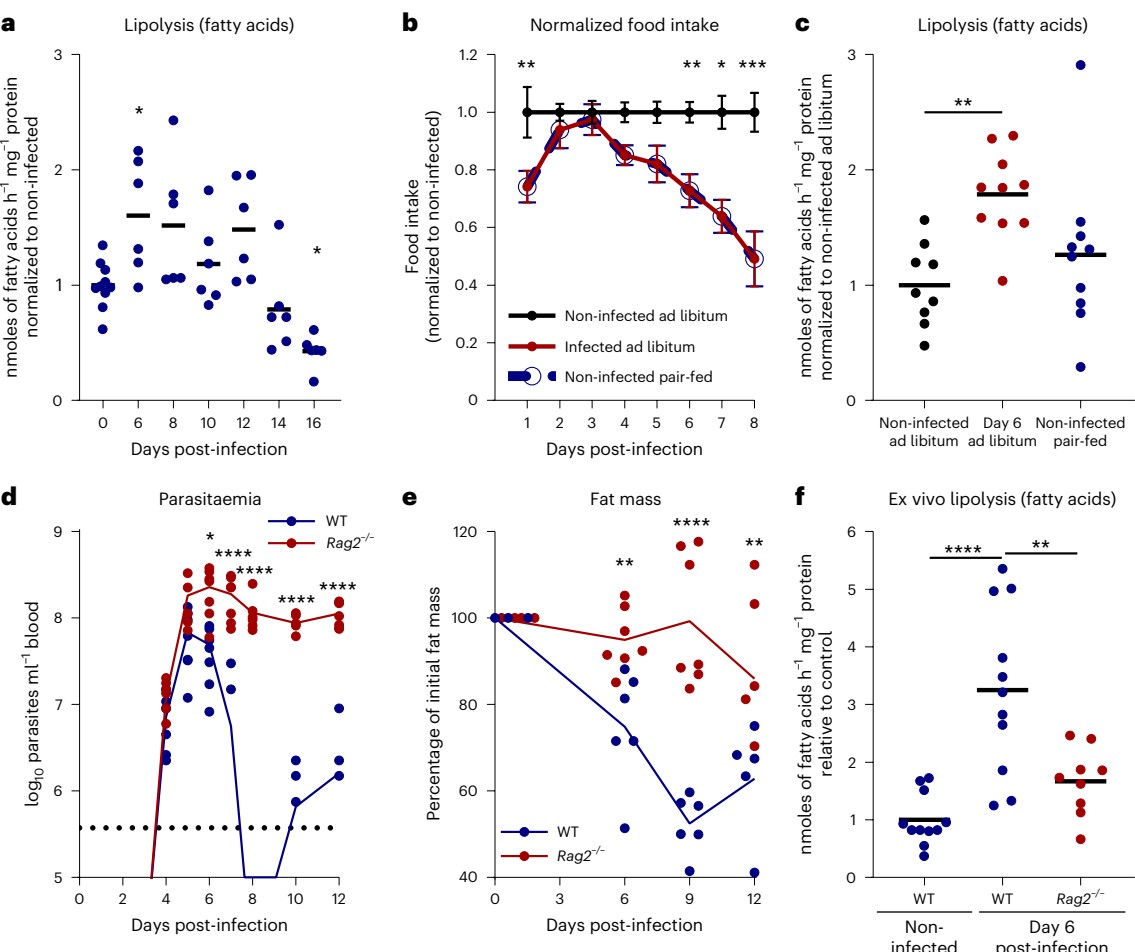

**Fig. 2 | Induced adipocyte lipolysis is independent of mice feeding behaviour and dependent on adaptive immune response. a**, Release of FFAs from gonadal AT explants of infected and non-infected WT mice under fasted and re-fed conditions, normalized to non-infected mice. For days 0 and 6–16, $n = 10$ and 6 mice, respectively, were examined in a single experiment. **b**, Daily food intake normalized to non-infected ad libitum-fed mice. For days 0–6 and 7–8, $n = 15$ and five mice were examined per group over two independent experiments. **c**, Release of FFAs from gonadal AT explants of infected and non-infected WT mice, normalized to ad libitum-fed non-infected mice. $n = 9$ mice for non-infected ad libitum group, and $n = 10$ for infected and non-infected pair-fed groups, examined over two independent experiments. **d**, Representative parasitaemia

of infected WT and $Rag2^{-/-}$ mice. $n = 8$ WT and 7 $Rag2^{-/-}$ mice examined in a single experiment. **e**, Fat mass relative to baseline of infected $Rag2^{-/-}$ and WT controls. $n = 6$ WT and 7 $Rag2^{-/-}$ mice examined over two independent experiments. **f**, Release of FFAs from gonadal AT explants of infected WT and $Rag2^{-/-}$ mice normalized to non-infected WT controls. $n = 11$ non-infected WT mice, 11 infected WT mice and 9 infected $Rag2^{-/-}$ mice examined over two independent experiments. Error bars represent the s.e.m. Statistical analysis was performed with one-way (**a**, **c** and **f**) and two-way (**b**, **d** and **e**) ANOVA using Šidák's test for multiple comparisons. *$P < 0.05$, **$P < 0.01$, ***$P < 0.001$, ****$P < 0.0001$. Statistical source data contain additional parameters.

of fat compared with $Atgl^{fl/fl}$ mice through all stages of infection (Fig. 3h). Consistently, necropsy of moribund $Adipoq^{Cre/+}$-$Atgl^{fl/fl}$ mice revealed significantly higher weight of gonadal AT depots than those of $Atgl^{fl/fl}$ mice (Extended Data Fig. 7b,c). These data suggest that both ATGL-dependent and ATGL-independent mechanisms contribute to fat loss during the full duration of a *T. brucei* infection.

Overall, these data indicate that ATGL activity in adipocytes during infection promotes loss of fat mass and potentially enables better lean mass retention.

**Lipolysis prolongs survival and reduces AT parasite burden**

Having established that adipocyte ATGL plays a central role in fat mass loss during *T. brucei* infection, next we asked how lipolysis affected disease progression. Interestingly, lipolysis-deficient mice succumbed earlier to infection than their WT littermate controls (Fig. 4a), suggesting a protective role for ATGL-mediated lipolysis during infection. This protection does not involve systemic parasite control, as both groups present similar parasitaemia profiles (Fig. 4b).

To understand how adipocyte lipolysis contributes to prolonged host survival, we questioned whether adipocyte lipolysis is able to modulate the number of AT-resident parasites per se. Quantification of parasite load in the gonadal AT (by quantitative polymerase chain reaction (qPCR)) revealed that the parasite number was ten-fold higher in lipolysis-deficient mice than in WT littermate controls at day 10 post-infection (Fig. 4c). By day 20 post-infection, this difference in parasite load was no longer detectable, which correlates with the fact that lipolysis is no longer activated at later stages of infection (Fig. 1). A similar profile was observed after normalization of parasite burden to tissue mass (that is, to remove bias due to loss of fat mass) (Extended Data Fig. 8). This higher parasite load appears to be restricted to the AT, as no significant differences in parasite load were found between the liver, spleen or lung of lipolysis-competent and lipolysis-deficient mice (Fig. 4d–f).

During *T. brucei* infection, the total parasite load depends on a quorum sensing mechanism that triggers differentiation of parasite replicative forms into non-replicative stumpy forms[26]. We hypothesised that the higher parasite load detected in gonadal AT of

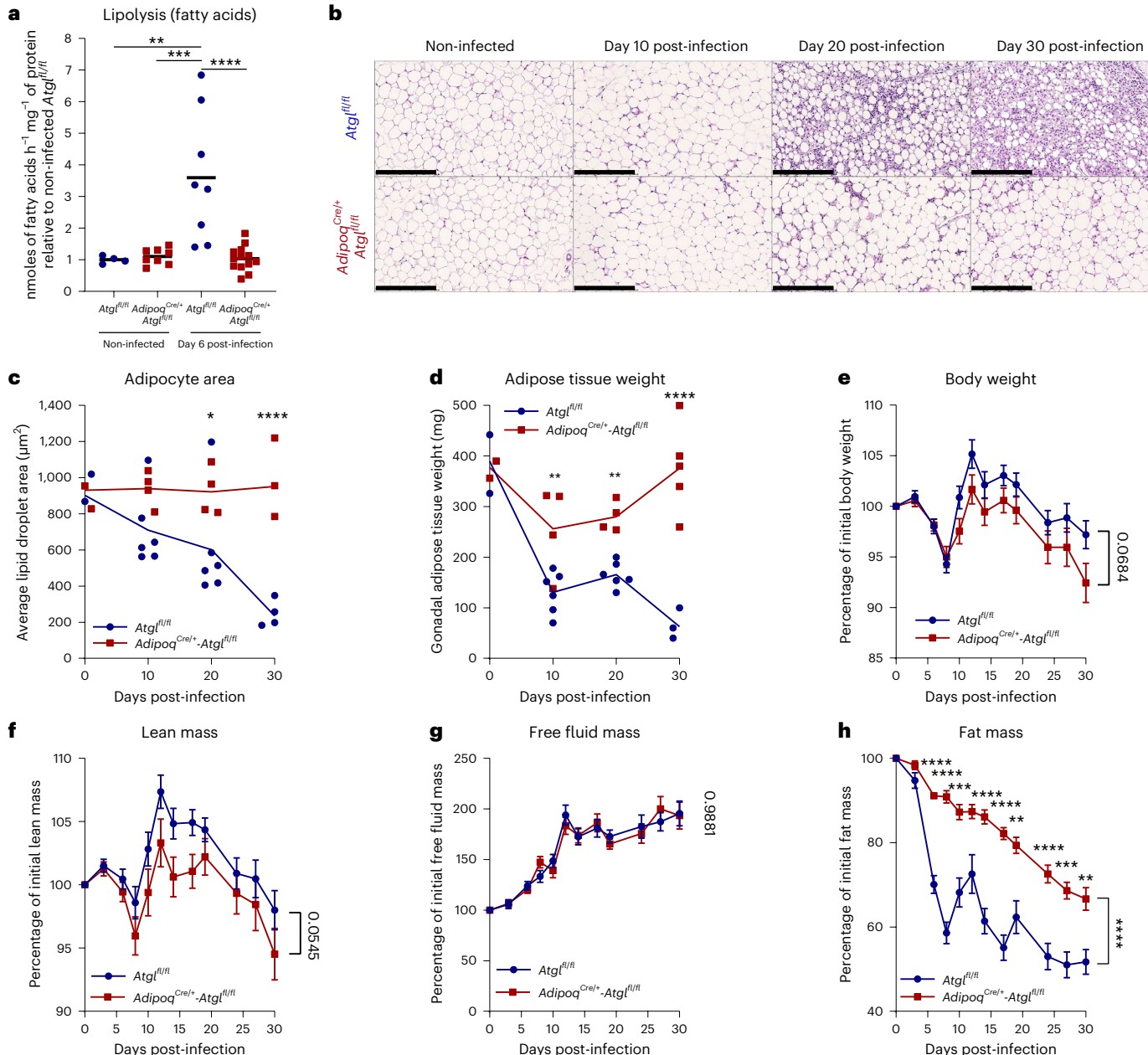

**Fig. 3 | ATGL deficiency in adipocytes prevents *T. brucei*-induced lipolysis and reduces global loss of fat mass. a**, Release of FFAs from gonadal AT explants of infected and non-infected *Atgl*^fl/fl (WT) and *Adipoq*^Cre/+-*Atgl*^fl/fl (KO) mice. *n* = 4 non-infected and 8 infected *Atgl*^fl/fl mice and 8 non-infected and 13 infected *Adipoq*^Cre/+-*Atgl*^fl/fl mice examined over two independent experiments. **b**, Representative gonadal AT haematoxylin and eosin micrographs (20× magnification; scale bar, 250 μm). **c**, Lipid droplet area of adipocytes in tissue sections of the gonadal AT. **d**, Total weight of the gonadal AT at different timepoints post-infection. For days 0, 10, 20 and 30, *n* = 3, 6, 6 and 5 *Atgl*^fl/fl mice and 3, 4, 4 and 5 *Adipoq*^Cre/+-*Atgl*^fl/fl mice were examined, respectively, in a single experiment (**c** and **d**). **e**–**h**, Total body weight (**e**), lean mass (**f**), free fluid mass (**g**) and fat mass (**h**) relative to baseline. *n* = 32 *Atgl*^fl/fl and 27 *Adipoq*^Cre/+-*Atgl*^fl/fl mice examined over four independent experiments (**e**–**h**). Error bars represent the s.e.m. Statistical analysis was performed with one-way ANOVA (**a**) or two-way ANOVA (**c** and **d**) or mixed-effects two-way ANOVA (**e**–**h**) using Šidák's test for multiple comparisons. *$P < 0.05$, **$P < 0.01$, ***$P < 0.001$, ****$P < 0.0001$. Statistical source data contain additional parameters.

lipolysis-deficient mice could be due to reduced parasite differentiation into the non-replicative stumpy form. To quantify the proportion of replicative and non-replicative forms in the AT population, we infected lipolysis-competent and lipolysis-deficient mice with a parasite reporter cell line that expresses green fluorescent protein (GFP) when differentiation is triggered (GPF::PAD1_{3'UTR} reporter). During the peak of gonadal AT parasite burden, a higher frequency of GPF::PAD1_{3'UTR} expression was found in parasites extracted from the AT of lipolysis-deficient mice (Fig. 4g), suggesting that in the absence of ATGL-mediated adipocyte lipolysis parasites still undergo differentiation.

Together, these data suggest that adipocyte lipolysis induces a local effect that limits the number of parasites, and this is associated with increased survival of the host.

### FFAs are cytotoxic to parasites

Next we questioned whether the differences in parasite burden observed in the gonadal AT between lipolysis-competent and lipolysis-deficient hosts could be because of a trypanotoxic accumulation of extracellular FFAs. To address this question, we co-cultured 3T3-L1 adipocytes and *T. brucei*.

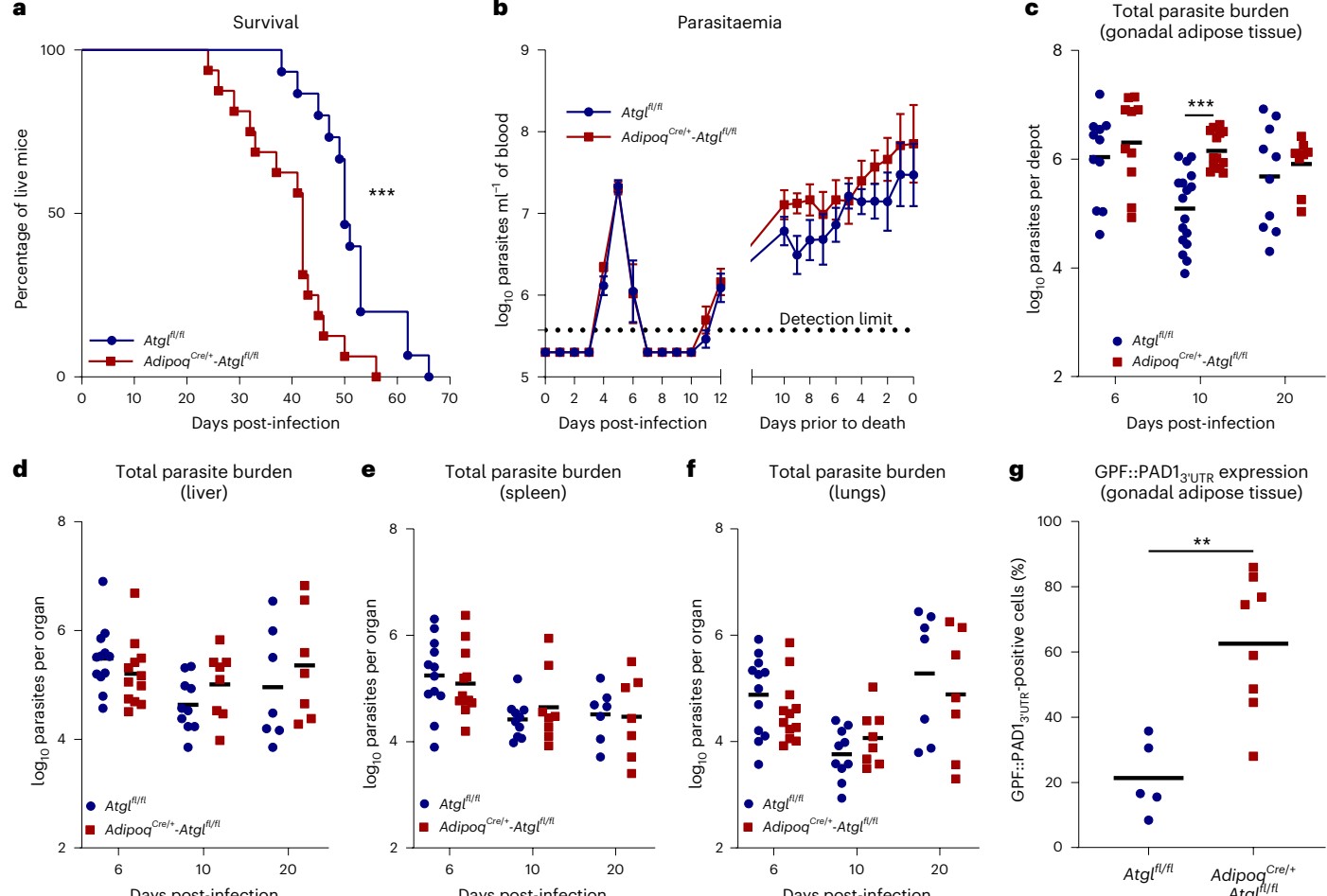

**Fig. 4 | Adipocyte-specific ATGL deficiency promotes reduction of host survival and enhancement of parasite load in AT. a**, Survival curves of *T. brucei*-infected *Atgl*^fl/fl^ and *Adipoq*^Cre/+^-*Atgl*^fl/fl^ mice. $n = 15$ *Atgl*^fl/fl^ and 16 *Adipoq*^Cre/+^-*Atgl*^fl/fl^ mice examined over two independent experiments. **b**, Representative early and terminal log~10~ parasitaemia of infected *Atgl*^fl/fl^ and *Adipoq*^Cre/+^-*Atgl*^fl/fl^ mice. $n = 9$ mice per group. **c**–**f**, log~10~ number of *T. brucei* parasites at different timepoints post-infection per gonadal (**c**), liver (**d**), spleen (**e**) and lungs (**f**) AT depots of *Atgl*^fl/fl^ and *Adipoq*^Cre/+^-*Atgl*^fl/fl^ mice. In **c**, for days 6, 10 and 20, $n = 11, 17$ and 10 *Atgl*^fl/fl^ mice and 10, 14 and 8 *Adipoq*^Cre/+^-*Atgl*^fl/fl^ mice, respectively, examined over two independent experiments. In **d**–**f**, for days 6, 10 and 20, $n = 12, 10$ and 7 *Atgl*^fl/fl^

mice and 12, 8 and 7 *Adipoq*^Cre/+^-*Atgl*^fl/fl^ mice examined over two independent experiments. **g**, Percentage of GFP::PAD1~3′UTR~ positive parasites extracted from the gonadal AT of *Atgl*^fl/fl^ and *Adipoq*^Cre/+^-*Atgl*^fl/fl^ mice at day 6 post-infection. $n = 5$ *Atgl*^fl/fl^ and 8 *Adipoq*^Cre/+^-*Atgl*^fl/fl^ mice examined in a single experiment. Error bars represent the s.e.m. Statistical analysis was performed with log-rank (Mantel–Cox) test (**a**), two-way ANOVA using Šidák's test for multiple comparisons (**b**–**f**) and two-sided unpaired *t*-test (**g**). *$P < 0.05$, **$P < 0.01$, ***$P < 0.001$. Detection limit is $3.75 \times 10^5$ parasites ml$^{-1}$ (**b**). Statistical source data contain additional parameters.

Daily treatment of 3T3-L1 adipocytes with forskolin led to a significant accumulation of FFAs in the supernatant, ranging from 0.34 mM to 0.79 mM over 3 days of incubation (Fig. 5a). Consistent with the hypothesis that FFAs are cytotoxic, parasites co-cultured with adipocytes in the presence of forskolin showed no substantial growth for 72 h after inoculation, while in the same period parasites in control co-cultures grew 35-fold (Fig. 5b). No direct effect of forskolin on parasite growth was observed in axenic cultures (Fig. 5c), indicating that the inhibitory effect of forskolin is mediated by adipocytes. Moreover, and as previously observed in parasites isolated from AT (Fig. 4f), increased adipocyte lipolysis in vitro did not lead to an increase in GPF::PAD1~3′UTR~ positive parasites (Fig. 5d), nor an increase in the proportion of parasites in G0/G1 cell cycle stage (Fig. 5e,f), suggesting that lipolysis does not enhance differentiation of slender to stumpy forms. It is intriguing that, upon forskolin stimulus, despite the almost absence of parasite growth, we can still detect a substantial proportion of parasites that have initiated differentiation, suggesting that in these conditions, differentiation may be less dependent upon parasite density.

To investigate whether *T. brucei* parasites might be subjected to cytotoxic amounts of FFAs in vivo, we quantified their abundance and species distribution in the AT interstitial fluid using gas chromatography–mass spectrometry (GC–MS). Both at day 6 post-infection and in non-infected mice, the most abundant FFA species found were (in order): palmitic acid (C16:0), linoleic acid (C18:2), oleic acid (C18:1), stearic acid (C18:0), myristic acid (C14:0) and palmitoleic acid (C16:1) (Fig. 6a). No significant differences in total FFA concentration were observed between non-infected and infected mice (4.8 mM versus 4.65 mM), suggesting that the upregulation of lipolysis does not cause a net increase in the global concentration of the most abundant FFAs.

Nevertheless, the relative distribution of interstitial FFAs showed significant changes during infection. Specifically, infected mice showed 52%, 20% and 56% reductions of C14:0, C16:0 and C16:1, respectively, while presenting a 40% increase in the quantity of both C18:0 and C18:1. Of the most abundant FFAs quantified, only C18:2 showed no differences between infected and non-infected mice.

Using the compositional information obtained from the interstitial fluid of infected mice we created supplemented mimetic medium

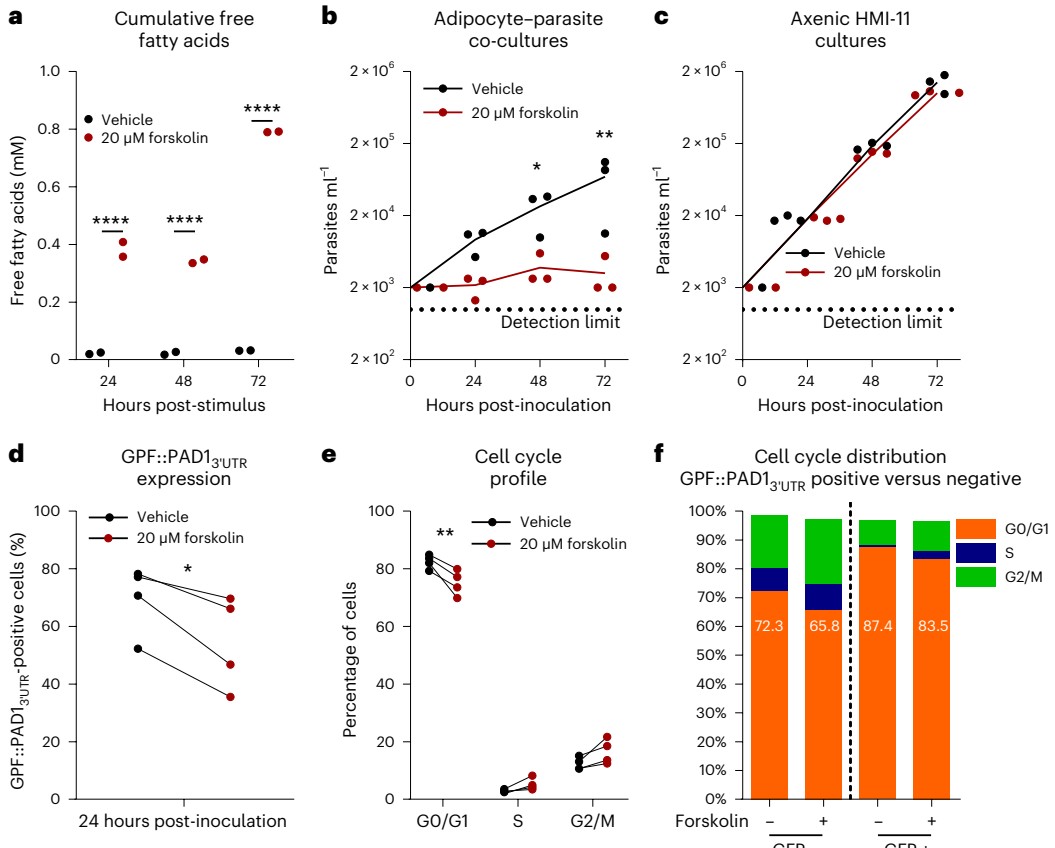

**Fig. 5 | Adipocyte lipolysis induction reduces parasite proliferation.**
**a**, Cumulative concentration of FFAs in culture medium in 3T3-L1 cultures upon daily stimulation with forskolin or vehicle. **b,c**, Parasite numbers observed in 3T3-L1 adipocyte co-cultures (**b**) and axenic HMI-11 cultures (**c**) in the presence of forskolin or vehicle (DMSO). **d,e**, Percentage of GFP::PAD1$_{3'UTR}$ positive (**d**) and cell cycle prolife (**e**) of parasites in 3T3-L1 co-cultures in the presence of forskolin

or vehicle at 24 h post-inoculation. **f**, Cell cycle distribution within GFP::PAD1$_{3'UTR}$ positive and GFP::PAD1$_{3'UTR}$ negative populations at 24 h post-inoculation. Data analysed using two-way ANOVA using Šidák's test for multiple comparisons (**a**–**c** and **e**) or two-sided paired *t*-test (**d**). *$P < 0.05$, **$P < 0.01$, ****$P < 0.0001$. $n = 2$ (**a**), $n = 3$ (**b** and **c**) and $n = 4$ (**d**–**f**) independent experiments. Statistical source data contain additional parameters.

containing each individual FFA or the entire mixture, that is, bound to bovine serum albumin (BSA), and assessed its effect on the proliferation and survival of bloodstream and AT forms. A 4.65-mM FFA mixture resulted in complete loss of parasite viability after 24 h of incubation (Fig. 6b,c and Extended Data Fig. 9). An intermediate effect was also observed in the presence of 5 mM of 1-hexadecanol, a compound structurally related to FFAs, but unable to follow similar metabolic pathways, suggesting that the loss of parasite viability in the presence of 4.65 mM of FFA mixture is more likely due to an intracellular cytotoxic effect rather than a detergent-like effect. FFA toxicity was comparable between bloodstream forms and in vivo isolated AT forms (Fig. 6b,c). Interestingly, individual testing of each FFA revealed that physiological concentrations of C18:2 (1.18 mM), but not any other FFA, were sufficient to reduce *T. brucei* viability to the same extent as the complete mixture (Fig. 6b,c and Extended Data Fig. 9). Importantly, these trypanocidal conditions, that is, 1.18 mM of C18:2 and 4.65 mM of mixture, showed no acute cytotoxicity to either splenocytes or 3T3-L1 pre-adipocytes (Fig. 6d), suggesting that host cells are tolerant to these interstitial FFA conditions.

Given that the effect of ATGL-lipolysis in AT parasite burden were transient (Fig. 4c), we next investigated whether progression of infection would result in a decrease of interstitial FFAs and thus a more permissive environment for the parasite. Using a colorimetric assay to measure FFA concentration, we confirmed that by days 9, 16 and 30 post-infection, interstitial concentrations of FFAs showed a 57%, 71% and 54% reduction, respectively, compared with non-infected mice

(Fig. 6e). Finally, we tested whether the levels of interstitial FFAs were dependent upon adipocyte ATGL activity. For that, we measured FFA concentration in the interstitial fluid of animals infected for 6 days and non-infected controls. Lipolysis-deficient mice (*Adipoq*$^{Cre/+}$-*Atgl*$^{fl/fl}$) presented with a 48% reduction in FFA abundance relative to non-infected lipolysis-competent mice (*Atgl*$^{fl/fl}$) (Fig. 6f).

Together, these data show that the FFA composition of the interstitial environment of AT is cytotoxic for *T. brucei* and requires ATGL activity to be kept at critical levels.

## Discussion

In this work, we demonstrated that during *T. brucei* infection, adipocytes shrink due to ATGL-dependent lipolysis. Similar observations have been recently described by Redford et al., in a different ATGL-deficient mouse[27]. In other infections, ATGL has also been associated with fat mass reduction, as seen in full ATGL-deficient mice protected from fat loss induced by Lewis lung carcinoma or B16 melanoma cells[15]. In reverse, in a lymphocytic choriomeningitis virus infection model, *Adipoq*$^{Cre/+}$-*Atgl*$^{fl/fl}$ mice did not exhibit protection against weight loss, highlighting distinct roles of ATGL in viral and parasitic infections[28].

It is possible that other lipid-based pathways may play a role in fat wasting. However, the unchanged adipocyte size and higher fat mass retention in the absence of ATGL activity (Fig. 3) suggest that lipolysis is the primary pathway for *T. brucei*-induced fat mass loss. This dominance of increased adipocyte lipolysis over reduced lipogenesis in fat loss has also been reported in cancer-associated cachexia[29].

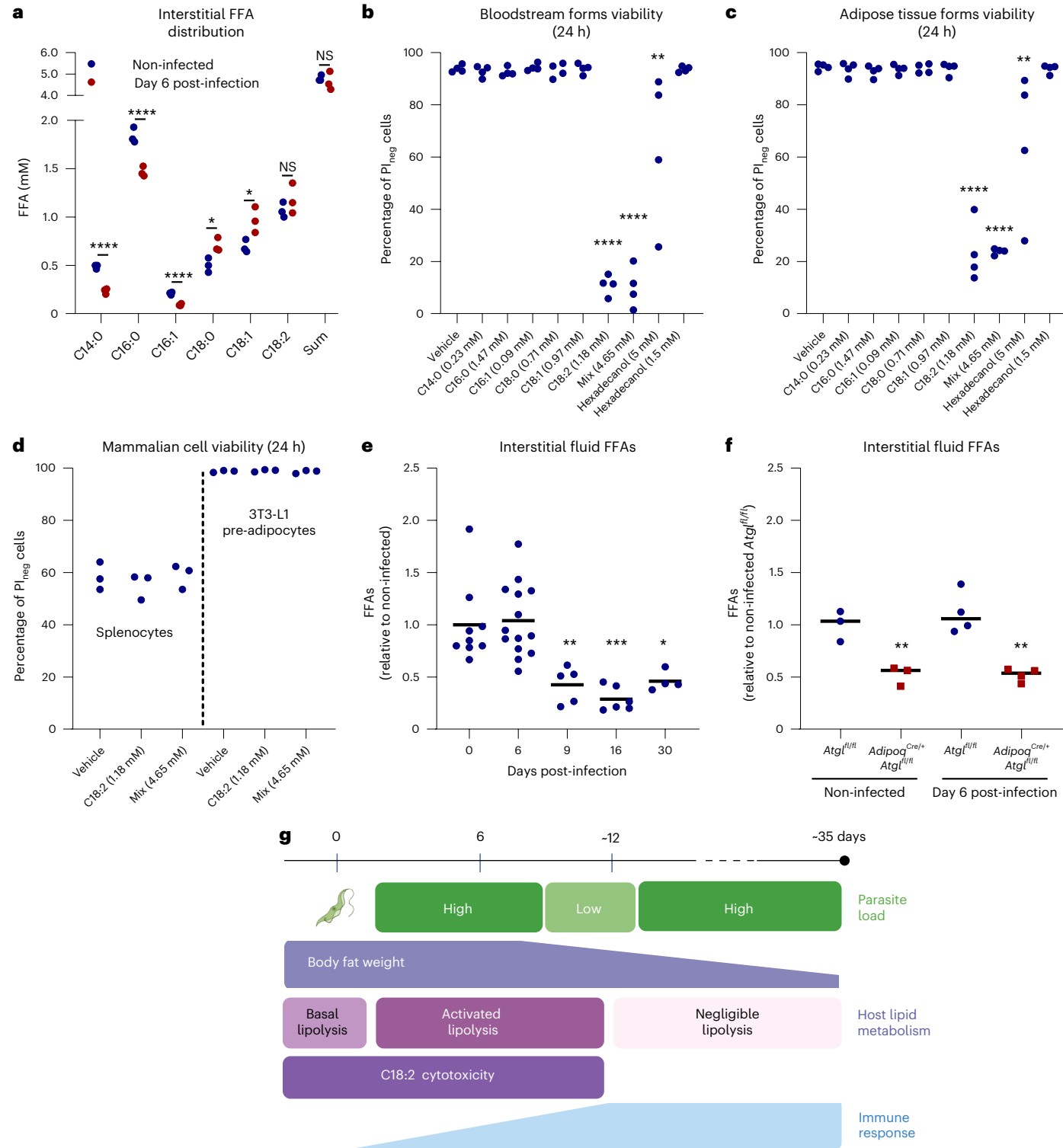

**Fig. 6 | AT interstitial FFAs are cytotoxic to *T. brucei*. a**, GC–MS quantification of the most abundant FFAs in the interstitial fluid collected from the gonadal AT of mice at day 6 post-infection and in non-infected controls. **b,c**, Viability of bloodstream form (**b**) or AT form (**c**) *T. brucei* parasites after 24 h of incubation with physiological concentrations of individual FFAs or with an in vivo mimetic FFA mixture. **d**, Viability of splenocytes and 3T3-L1 pre-adipocytes after 24 h of incubation with physiological concentrations of C18:2 acids or with an in vivo mimetic FFA mixture. **e,f**, Relative AT interstitial FFAs measured with a colorimetric assay in wild-type (**e**) and *Atgl*[fl/fl] or *Adipoq*[Cre/+]-*Atgl*[fl/fl] mice (**f**). **g**, Model displaying important changes in parasite number (green), host

lipid metabolism, in particular metabolism of FFAs (purple) and immune response (blue) (from ref. 7) at different stages of *T. brucei* mouse infection. *n* = 3 independent experiments using five mice per group per experiment (**a**). *n* = 4 (**b** and **c**) and *n* = 3 independent experiments (**d**). For days 0, 6, 9, 16 and 30, *n* = 9, 14, 5, 6 and 4 mice examined over two independent experiments (**e**). *n* = 3 non-infected and 4 infected *Atgl*[fl/fl] mice and 3 non-infected and 4 infected *Adipoq*[Cre/+]-*Atgl*[fl/fl] mice examined in a single experiment (**f**). Data analysed using multiple unpaired two-sided *t*-tests (**a**) and one-way ANOVA using Šidák's test for multiple comparisons (**b**–**f**). \**P* < 0.05, \*\**P* < 0.01, \*\*\**P* < 0.001, \*\*\*\**P* < 0.0001. NS, not significant. Statistical source data contain additional parameters.

The ablation of adipocyte ATGL did not completely prevent whole-organism fat mass loss during *T. brucei* infection (Fig. 6d). This indicates that other mechanisms, such as TNF-α-mediated lipoprotein lipase inhibition may lead to reduced lipogenesis and contribute to fat mass reduction[30]. Despite the absence of adipocyte lipolysis, a substantial fat mass decrease would still be expected due to the capacity of most cells to store fats in cytosolic lipid droplets. Additionally, in the event of activated acid lipolysis in adipocytes during infection, lysosomal acid lipase would hydrolyse TAGs into FFAs and hydrolyse glycerol independently of neutral lipases such as ATGL[31].

To identify the signal enhancing lipolysis, we investigated specific factors inducing adipocyte lipolysis during *T. brucei* infection. Our findings suggest that changes in sympathetic tone or feeding behaviour are insufficient to drive *T. brucei*-induced adipocyte lipolysis (Fig. 2a–c and Extended Data Fig. 5a). Interestingly, our observations differ from Redford et al., who reported that anorexia alone is sufficient to induce adipose wasting[27]. This difference may be attributed to the severity of their infection model, as they injected mice with 500,000 parasites compared with our 2,000 parasites.

We found that T and B cell-secreted cytokines, or those secreted by other cells upon T and/or B cell-dependent activation, are more likely to be the inducers of adipocyte lipolysis (Fig. 2e,f). Numerous immune factors, particularly cytokines, have been shown to have a pro-lipolytic effect[32,33]. Notably, a recent study found that interleukin 17 signalling drives AT weight loss and limits *T. brucei* parasite burden in subcutaneous AT[34]. In another investigation, CD4+ T cells were identified as necessary for the sickness-induced anorexic response, leading to ATGL activation in adipocytes, which is essential for the occurrence of wasting[27]. Given the complex inflammatory nature of *T. brucei* infection, it is plausible that multiple cells and factors are simultaneously elevated, exerting a combined/synergistic pro-lipolytic effect.

Blocking ATGL-mediated lipolysis does not affect parasitaemia, but it leads to a ten-fold increase in the number of parasites in AT compared with when lipolysis is active. We showed that inhibiting lipolysis did not impair parasite differentiation in AT (Fig. 4g). Similarly, promoting lipolysis did not enhance parasite differentiation (Fig. 5d–f). Instead, we found that adipocyte lipolysis reduces the number of parasites by releasing cytotoxic FFAs. Importantly, basal ATGL activity was sufficient to maintain these critically elevated FFA levels (Fig. 6a,f), suggesting that local FFA carrier proteins are at or near maximum capacity under steady-state conditions. It is not unplausible that, when lipolysis is stimulated, newly released FFAs rapidly flux through the endothelium into the bloodstream, get recycled into adipocytes or are consumed by neighbouring cells. This environment, toxic to trypanosomes, exists in the AT of healthy mice and at least during the initial 6 days of infection. However, this FFA-driven cytotoxicity diminishes later in the infection, as adipocytes undergo atrophy and potentially lose their ability to hydrolyse enough TAG to replenish interstitial FFA levels (Fig. 6e and Extended Data Fig. 4).

Although AT forms display enhanced capacity to catabolize some FFAs (C14:0)[5], we discovered that they are susceptible to physiological concentrations (-1.18 mM) of C18:2. One possibility is that excessive endocytosis of C18:2 by *T. brucei* could collectively exceed its capacity to catabolize these species through β-oxidation[5], or to incorporate them into stable acylglycerol species stored in lipid droplets or utilized within membrane phospholipids. A similar lipid-mediated killing has been demonstrated in the intracellular parasite *Toxoplasma gondii*, where excessive supplementation of unsaturated C18 and C16 FFAs or inhibition of *T. gondii* diacylglycerol acyltransferase 1, that is, inhibiting lipid droplet formation, led to a marked increase in doubling time and parasite viability[35]. Defective lipid droplet formation in the *T. brucei* insect stage (procyclic form) leads to mitochondrial abnormalities and reduced parasite viability[36]. Hence, it is plausible that similar lipotoxicity phenotypes may occur with mammalian stages of *T. brucei* in the interstitial spaces of the AT.

Our previous studies showed that both the pancreas and AT serve as major parasite reservoirs[5,6]. In the AT, the parasites adapt their gene expression, catabolize C14:0 (ref. 5) and reduce growth rate[37]. These results supported a model in which globally the AT was a favourable environment for *T. brucei*. However, the current study demonstrates that the situation is more complex, as adipocyte lipolytic activity, present in healthy mice and upregulated during the first 10 days of infection, is actually detrimental to a fraction of the parasite population (Fig. 6g). Despite the cytotoxicity of C18:2 to the parasites, a substantial number (between $10^5$ and $10^6$) of parasites are still detectable in the AT of WT animals (Fig. 4c). This suggests that either a subpopulation of parasites is resistant to the toxic effects, and/or there is a constant influx of parasites from the bloodstream replenishing the AT. The influx of parasites is probably predominant early in infection because parasitaemia is also growing exponentially in the blood, peaking between days 5 and 6. Thus, the net effect of lipid toxicity might be relatively minor during the initial 6 days of infection. As the infection progresses and adipocytes deplete their TAG storage, lipolysis ceases and lipid cytotoxicity is no longer expected, potentially making other parasite control mechanisms, such as the immune response, more dominant (Fig. 6g). Indeed, earlier studies have shown higher numbers of immune cells in AT later in infection compared with day 6 (ref. 7), suggesting that the immune response probably plays a more substantial role in controlling parasite load at this stage.

In this work, we focused on the role of FFAs. Future studies will be necessary to dissect the role of glycerol, the other end product of lipolysis. Glycerol concentration in vivo depends not only on host lipid metabolism (including lipolysis), but also on secretion/consumption by parasites. Our results are consistent with previous studies showing that *T. brucei* produces glycerol, and this production is inversely proportional to the concentration of oxygen[38,39]. Thus, it is likely that within tissues, parasites secrete more glycerol than in the blood. Another confounding factor is that high concentrations of glycerol can trigger differentiation of bloodstream form parasites into stumpy forms in vitro[40] suggesting that this metabolite may have a role in parasite differentiation. Careful tracing experiments will be necessary to disentangle the origin and consequences of glycerol metabolism during a *T. brucei* infection.

Neutral lipolysis in AT is a protective mechanism against *T. brucei* infection. While not statistically significant, lipolysis-deficient mice exhibit a clear trend of lower total body weight and lean mass compared with lipolysis-competent mice. This suggests that in the absence of efficient adipocyte lipolysis, mice experience a more severe wasting phenotype, contributing to premature death. Further studies are necessary to fully understand the global physiological implications of ATGL-mediated lipolysis during *T. brucei* infection.

In conclusion, this study reveals that C18:2, a prevalent FFA in the interstitial space of AT, is toxic to *T. brucei* parasites and contributes to controlling parasite load in this tissue, as long as ATGL can access stored TAG and release FFAs into the interstitial fluid. In future studies, it will be intriguing to investigate how this cytotoxicity operates in *T. brucei*, whether similar processes occur in other lipid-rich organs, such as skin and pancreas, and in other infections caused by pathogens that colonize the AT (such as *Trypanosoma cruzi* and *Plasmodium*)[41]. Understanding these mechanisms may offer valuable insights into host–parasite interactions and potential therapeutic strategies against such infections.

## Methods
### Ethics statement
Animal experiments were performed according to European Union regulations and approved by the Órgão Responsável pelo Bem-estar Animal (ORBEA) of Instituto de Medicina Molecular João Lobo Antunes and the competent authority Direcção Geral de Alimentação e Veterinária (licences 018889\2016 and 017549\2021).

## Animals

C57BL/6J mice were purchased from Charles River Laboratories International. Genetically modified mice include: $Rag2^{-/-}$ (IMSR_JAX:008449), $Tnfa^{-/-}$ (IMSR_JAX:005540), $Adipoq^{Cre/+}$ (IMSR_JAX:010803)[23] and $Atgl^{fl/fl}$ (IMSR_JAX:024278)[24]. All experimental C57BL/6J WT, $Rag2^{-/-}$ and $Tnfa^{-/-}$ mice were males between 8 and 12 weeks old. Sex- and age-matched $Adipoq^{Cre/+}Atgl^{fl/fl}$ were used aged between 8 and 20 weeks old. Mice were housed in a specific pathogen-free barrier facility, under standard laboratory conditions: 21–22 °C ambient temperature, a 12-h light/12-h dark cycle and 45–65% humidity. Chow and water were available ad libitum, unless otherwise stated.

## Infection

*T. brucei* cryostabilates were thawed and parasite viability was confirmed by observing motility under an optic microscope. Mice were infected by intraperitoneal injection of 2,000 *T. brucei* parasites. At selected timepoints post-infection, animals were euthanized by $CO_2$ narcosis and immediately perfused transcardially with pre-warmed heparinized saline (50 ml PBS with 250 µl of 5,000 IU ml⁻¹ heparin).

## Feeding synchronization

Feeding synchronization was achieved by fasting mice for 7 h from 0:00 to 7:00. Afterwards, access to food was re-established for 2 h, after which mice were euthanized.

## Paired feeding

Mice were individually housed, and their food intake monitored daily. To each infected mouse a non-infected control was assigned, that is, a pair-fed control. Every 24 h, pair-fed controls were provided with the same amount of food consumed by their respective infected pair in the previous 24 h.

## Body composition analysis

Mouse total fat mass, lean mass and free body fluid mass were determined using a 6.2-MHz time-domain nuclear magnetic resonance-based Minispec LF65 (Bruker) apparatus with the Minispec Plus (version 7.0.0) software. Live unanaesthetized mice were restrained, weighed and then inserted into the Minispec apparatus. Each measurement lasted for approximately 1 min.

## Chemical sympathectomy

Chemical sympathectomy was performed by intraperitoneal administration of 200 mg kg⁻¹ of 6-OHDA in PBS containing 0.4% ascorbic acid as stabilizer. A total of 72 h and 24 h before infection, mice were treated with either 6-OHDA or with PBS containing 0.4% ascorbic acid (sham controls)[42].

## Parasite lines

Mouse and in vitro infections were performed using parasites derived from *T. brucei* AnTat 1.1E, a pleomorphic clone derived from the EATRO1125 strain. AnTat 1.1E 90–13 is a transgenic cell line encoding the tetracyclin repressor and T7 RNA polymerase. AnTat 1.1E 90–13 GPF::PAD1₃′UTR reporter derives from AnTat 1.1E 90–13, in which the GFP is coupled to PAD1 3′ untranslated region (UTR). All parasite cell lines were propagated and maintained in HMI-11 medium at 37 °C in a 5% $CO_2$ atmosphere[43].

## 3T3-L1 cell culture

Confluent murine 3T3-L1 pre-adipocytes (CL-173; ATCC) were differentiated in accordance with a previously described method[44] using 4.5 g l⁻¹ glucose Dulbecco's modified Eagle medium (DMEM) supplemented with GlutaMAX and containing 1 µM dexamethasone (Sigma, D4902), 0.5 mM 3-isobutyl-1-methylxanthine (Sigma, I7018), 1 µg ml⁻¹ insulin (Sigma, I9278) and 2 µM rosiglitazone (Santa Cruz, sc-202795). Cells were then kept in 4.5 g l⁻¹ glucose DMEM supplemented with GlutaMAX,

pyruvate and 10% foetal bovine serum, which was refreshed every 48 h until cells were used. 3T3-L1 adipocytes were inoculated with $2 \times 10^3$ parasites ml⁻¹ and stimulated with 20 µM of forskolin or dimethylsulfoxide (DMSO) every 24 h. The 3T3-L1 co-cultures were performed using DMEM containing 10% foetal bovine serum.

## Ex vivo lipolysis assay

Following euthanasia and systemic re-perfusion as previously described, AT depots were collected, rinsed with PBS and kept at 37 °C in low-glucose (1 g l⁻¹) DMEM (Gibco, Thermo Fisher Scientific) without serum until processing. Next, AT depots were cut into ~20-mg explants and incubated for 2 h at 37 °C in 96-well plates containing 200 µl low-glucose DMEM with 5% (w/v) FFA-free BSA and 5 µM of Triacsin C (Sigma, T4540) per well. Afterwards, AT explants were incubated for an additional hour with a similar medium containing 20 µM of forskolin (Abcam, ab120058). Up to five explants were used per AT depot. Next, FFA (C8 and longer) and glycerol concentrations were assessed using commercially available colorimetric kits (MAK044 and MAK117, Sigma) according to the manufacturer's instructions. Concentration of FFAs and glycerol was normalized to each explant's protein content. Protein contents were quantified by first delipidating the explants for 1 h in 1 ml of 2:1 chloroform–methanol and 1% acetic acid solution at 37 °C under vigorous agitation on a benchtop thermomixer. Afterwards, delipidated explants were lysed overnight in 500 µl of a 0.3 M NaOH and 0.1% (w/v) sodium dodecyl sulfate solution at 56 °C under vigorous agitation. Lysate protein quantification was performed using a BCA Protein Assay Kit (Thermo Fisher Scientific, A53225) according to the manufacturer's instructions. Ex vivo lipolysis rates are expressed as: nanomoles of metabolite released per milligram or protein per hour. Optical density measurements were performed in an Infinite M200 plate reader (Tecan) using i-control software (Tecan, version 2.0).

## Adipocyte lipid droplet area quantification

Images of haematoxylin and eosin-stained slides of gonadal AT were acquired in a Nanozoomer-SQ (Hamamatsu Photonics) using NDP. scan (version 1.0) with a magnification of 20×. Afterwards, at least five random fields per slide were analysed with ImageJ (version 1.51 h) using the Adiposoft (version 1.16) plugin[45] using an adipocyte diameter range from 10 to 100 µm.

## Parasite quantification in blood, organs and cultures

For parasitaemia quantification, blood samples were taken daily from the tail vein and diluted 1:150 in a PBS solution containing 2% formaldehyde. Parasites were counted manually in disposable 0.1 mm depth counting chambers (Kova International). Parasitaemia detection limit is $3.75 \times 10^5$ parasites ml⁻¹ of blood (equivalent to one parasite counted in a total of four squares with 0.1 µl in each). Determination of parasite numbers in axenic cultures or 3T3-L1 co-cultures was done by counting motile parasites in undiluted samples loaded onto disposable counting chambers. Detection limit for parasites under culture conditions is $2 \times 10^3$ (equivalent to one parasite counted in a total of ten squares).

For parasite quantification in organs, genomic DNA (gDNA) was extracted using NZY tissue gDNA isolation kit (NZYTech, Portugal). The amount of *T. brucei* 18S rDNA was measured by qPCR, using the primers 5′-ACGGAATGGCACCACAAGAC-3′ and 5′-GTCCGTTGACGGAATCAACC-3′, and converted into number of parasites using a calibration curve, as previously described[5]. Assays were run on QuantStudio 5 real-time PCR system (Thermo Fisher) using the firmware version 1.5.1 and analysed using QuantStudio Design and Analysis software version 2.6. Number of parasites per mg of organ (parasite density) was calculated by dividing the number of parasites by the mass of organ used for qPCR. The total amount of parasites in the organ was estimated by multiplying parasite density by the total mass of the organ.

## Preparation of single-cell suspensions

Single-cell suspensions containing AnTat 1.1E 90−13 GFP::PAD1$_{3'UTR}$ parasites were prepared for analysis of PAD1 expression by flow cytometry. Parasite AT forms were recovered by incubating the gonadal AT of infected mice in 5 ml of HMI-11 medium within 50 ml conical tubes under gentle agitation at 37 °C for 30 min. Afterwards, cell suspensions were centrifuged at 770*g*, washed with PBS, fixed with 2% formaldehyde for 20 min and washed again with PBS.

Splenocytes were obtained by sieving a spleen through a 40-μm pore size nylon mesh (BD Biosciences) with a syringe plunger. Cell suspensions were treated with red blood cell lysis buffer (BioLegend 420301).

## AT interstitial fluid isolation and FFA extraction

Interstitial fluid was collected as previously described[46]. Briefly, gonadal fat pads were excised, mounted on a 20-μm nylon net (NY2004700, Merck Life Sciences), placed on a 1.5-ml collection tube and centrifuged for 10 min at 800*g* at 4 °C. For subsequent GC−MS analysis, each sample was mixed in a 1:1 ratio with a 1 mM heptadecanoic acid (H3500, Merck Life Sciences) methanol solution (that is, internal standard) and then flash frozen until further processing. FFAs were then extracted using a variation of Dole's protocol[47]. Samples were mixed with water to a total volume of 0.5 ml, vortexed for 30 s, mixed with 2.5 ml of 80/20 2-propanol/n-hexane (v/v) with 0.1% $H_2SO_4$ and vortexed for 30 s. A total of 1.5 ml of *n*-hexane and 1 ml of water were then added to each sample, vortexed for 30 s and allowed to separate into phases. The supernatant was collected and evaporated in a glass vial, and the dry residue was used for downstream GC−MS analysis.

## FFA transmethylation and GC−MS analysis

Transmethylation of FFAs was performed on the dry total lipid extracts. The reaction (total volume 1 ml) is conducted in a glass vial. A total of 100 μl of toluene was added, followed by 750 μl of MeOH and 150 μl of 8% HCl MeOH:$H_2O$ 85:15 (v/v) solution to allow the esterification of the FFAs. The reaction was left to go to completion at 45 °C overnight. Upon drying, the fatty acid methyl esters (FAMEs) were extracted with a 1:1 *n*-hexane:$H_2O$. The FAME extracts were dried under nitrogen gas stream. The FAME extracts were dissolved in dichloromethane, typically 20 μl, and 1 μl was analysed by GC−MS on an Agilent Technologies GC-6890N gas chromatograph coupled to an MS detector-5973. Separation by GC was performed using a PhenomenexZB-5 column (30 M × 25 mm × 25 mm), with a temperature programme of 70 °C for 10 min, followed by a gradient to 220 °C, at 5 °C min$^{-1}$ and maintained at 220 °C for a further 15 min. Mass spectra were acquired from 50 to 500 a.m.u. The identity of FAMEs was carried out by comparison of the retention time and fragmentation pattern against bacterial and mammalian FAME standards and online available FAME library[48].

## Preparation of FFA solutions

Stock solutions of 10 mM sodium myristate (M8005), 25 mM sodium palmitate (P9767), 50 mM palmitoleic acid (P9417), 10 mM sodium stearate (S3381), 50 mM sodium oleate (O7501) and 50 mM linoleic acid (L8134), all from Merck Life Sciences, were prepared by dissolving each FFA in a 1:1 chloroform:methanol solution. Stock solutions were kept at −20 °C. To prepare working solutions, each stock was briefly warmed a 42 °C until all precipitates re-solubilized. Next, up to 500 μl of each stock solution was transferred to a chloroform compatible 5 ml tube and the organic solvent was allowed to fully evaporate. Vehicle controls were generated using the same process with a FFA-free 1:1 chloroform:methanol solution. Afterwards, FFAs were re-solubilized by first adding ethanol to re-suspend any residue on the tube's surface followed by warm HMI-11 medium containing 5% (w/v) FFA-free BSA, generating a solution containing 0.5% ethanol. Each mixture was then vortexed for 2 min and subsequently passed through a 0.22-μm filter. Each experiment was performed using freshly prepared working solutions.

## Flow cytometry

Cell suspensions containing formaldehyde-fixed AnTat 1.1E 90−13 GFP::PAD1$_{3'UTR}$ parasites extracted from infected gonadal AT depots were stained with a PBS solution containing 0.005 mg ml$^{-1}$ Hoechst 33342 (Thermo Fisher Scientific) for 20 min at 4 °C.

Viability analysis was performed by staining live AnTat 1.1E 90−13 parasites, splenocytes or 3T3-L1 pre-adipocytes in culture medium with 0.01 mg ml$^{-1}$ of propidium iodide (P4864, Merck Life Sciences).

For cell cycle analysis, 0.2−1 million cells were prepared as previously described[49] and then stained with 0.01 mg ml$^{-1}$ of propidium iodide.

Samples were passed through a 40-μm pore size nylon cell strainer (BD Biosciences) and then analysed on a BD LSRFortessa flow cytometer with FACSDiva 6.2 Software. All data were analysed using FlowJo software version 10.0.7r2. Schematics of the gating strategies used parasite analysis are represented in Extended Data Fig. 10.

## Statistical analysis

Statistical analysis was performed using GraphPad Prism (version 8.4.3). Data are presented as individual values or as mean ± standard error of the mean (s.e.m.). Parasite numbers were transformed into their respective log base 10 values to achieve linearization before statistical analysis. For analysis purposes, parasitaemia and culture parasite density values below the limit of detection were attributed a log$_{10}$ value equal to 5 and 2.69, respectively (that is, close the limit of detection). Statistical differences were assessed using two-way analysis of variance (ANOVA) and one-way ANOVA with Šidák's test for multiple comparisons. Stand-alone comparisons were performed using two-sided paired or unpaired *t*-tests. *P* values lower than 0.05 were considered to be statistically significant.

## Reporting summary

Further information on research design is available in the Nature Portfolio Reporting Summary linked to this article.

## Data availability

Statistical analysis for all main figures and extended data figures is provided in this article. Source data are provided with this paper.

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

## Acknowledgements

We are thankful to B. Silva-Santos and M. Veldhoen for providing experimental mice, and P. Scherer and the Figueiredo lab members for helpful discussions. We thank K. Matthews (University of Edinburgh) for providing the EATRO1125 AnTat 1.1E clone and C. Janzen (University of Wurzburg) for providing the GFP::PAD1$_{3'UTR}$ cell line. We are also grateful to the staff of iMM's rodent facility for excellent animal husbandry and welfare services and to the staff of iMM's Comparative Pathology Laboratory for expert technical

assistance. Illustrations were made using BioRender. This work was supported by European Research Council (ERC) under the European Union's Horizon 2020 research and innovation programme (grant agreement no. 771714), Fundação para a Ciência e Tecnologia (PD/BD/128286/2017 to H.M., CEECINST/00110/2018 to L.M.F.), the Austrian Fonds zur Förderung der Wissenschaftlichen Forschung (F73 SFB Lipid Hydrolysis to R.Z.) and the Louis Jeantet Prize 2015 by the Fondation Louis Jeantet to R.Z.

## Author contributions

H.M., P.H., R.Z., T.K.S. and L.M.F. designed the experiments. H.M. performed and analysed the experiments shown in Figs. 1–6 and Extended Data Fig. 1–10. P.H. and R.Z. provided technical assistance for experiments shown in Fig. 2a–c and Extended Data Fig. 6. T.K.S. performed the gas chromatography and mass spectrometry measurements and analysis shown in Fig. 6a. H.M. and L.M.F. wrote the manuscript with input from all co-authors. L.M.F. supervised the work.

## Competing interests

The authors declare no competing interests.

## Additional information

**Extended data** is available for this paper at https://doi.org/10.1038/s41564-023-01496-7.

**Correspondence and requests for materials** should be addressed to Luísa M. Figueiredo.

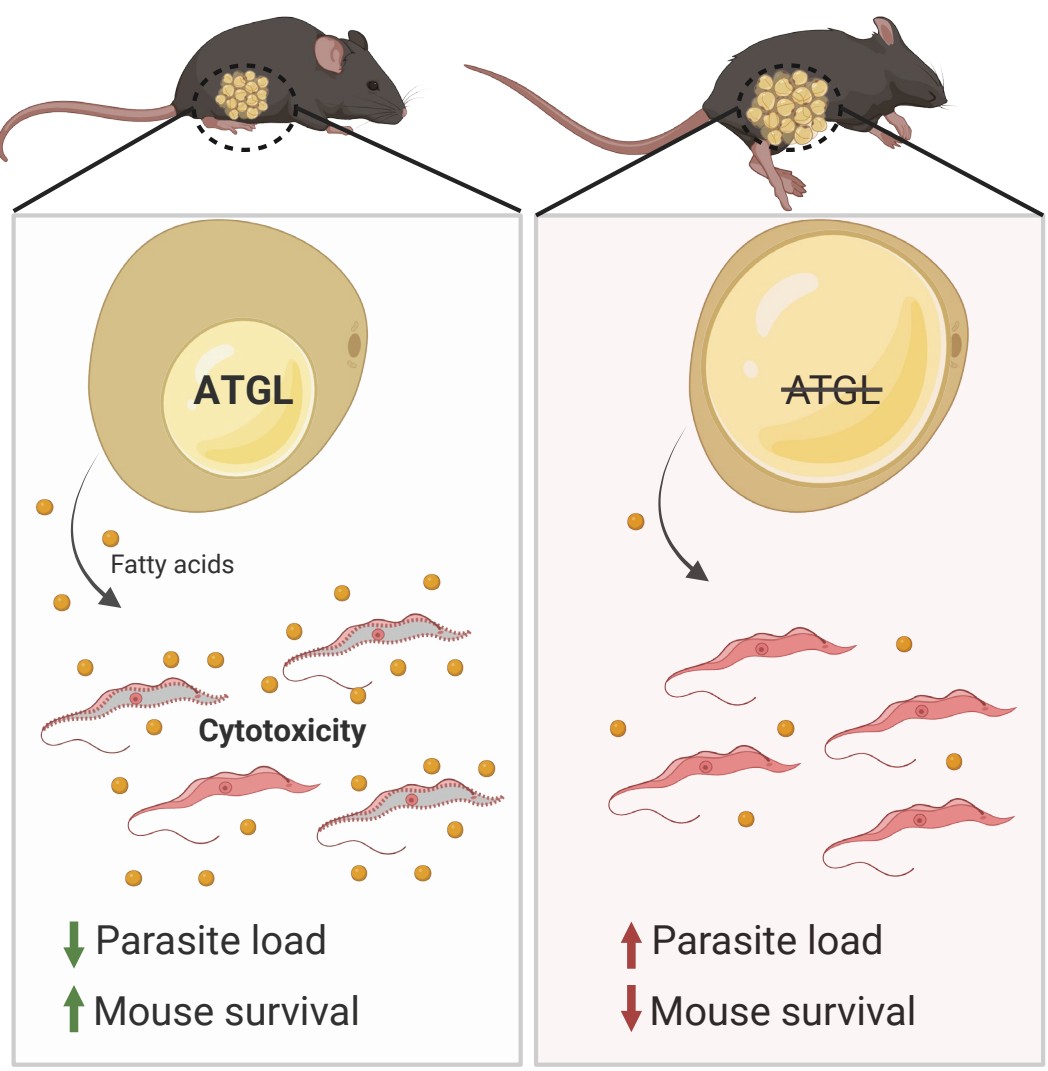

**Extended Data Fig. 1 | Protective effect of adipocyte lipolysis in murine *T. brucei* infection.** Graphical representation of the effects of ATGL-dependent lipolysis. ATGL activity promotes increased FFA release from adipocytes, leading to a reduction in adipocyte/AT size and maintaining a cytotoxic extracellular environment for parasites while prolonging mouse survival.

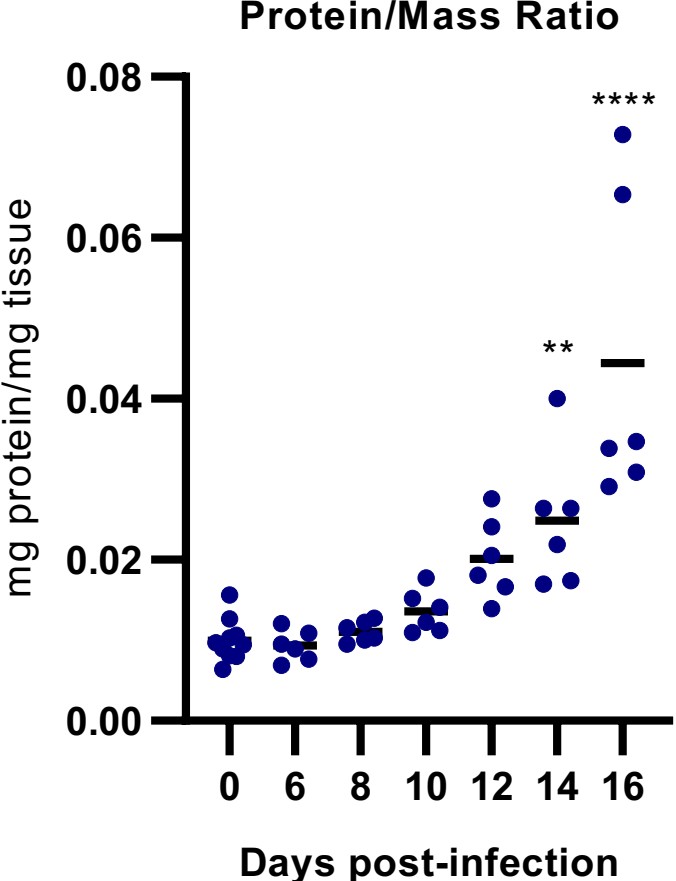

**Extended Data Fig. 2 | Dynamics of gonadal AT mass to protein ratio.** Gonadal AT explant protein to total mass ratio. Statistical analysis was performed with One-way ANOVA using Sidak's test for multiple comparisons. **, P < 0.01; ****, P < 0.0001. n(day 0; 6-16) = 10 and 6 mice examined in a single experiment. Statistical source data contains additional parameters.

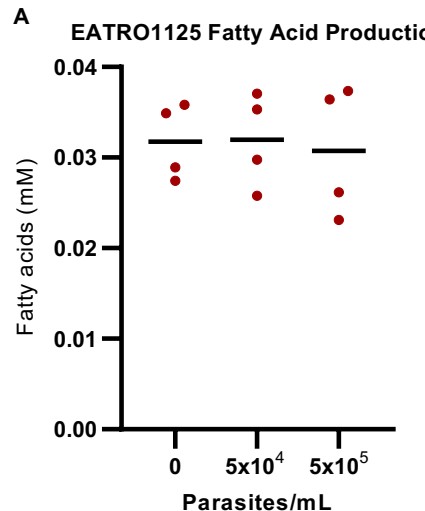

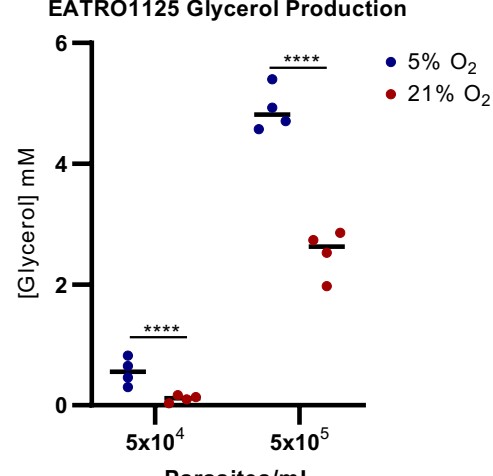

**Extended Data Fig. 3 | Parasite secretion of lipolytic metabolites.**
Concentration of **(a)** FFAs in axenic HMI-11 cultures with 5% (w/v) BSA after
24 hours of incubation without parasites or with an initial inoculum of $5 \times 10^4$ or
$5 \times 10^5$ parasites/mL. Concentration of **(b)** glycerol in axenic HMI-11 cultures after
24 hours of incubation an initial inoculum of $5 \times 10^4$ or $5 \times 10^5$ parasites/mL under

5% or 21% oxygen levels. Statistical analysis was performed with **(a)** one-way
ANOVA using Sidak's test for multiple comparisons and **(b)** two-sided unpaired
t-test. ***, $P < 0.001$; ****, $P < 0.0001$. $n = 4$ independent experiments. Statistical
source data contains additional parameters.

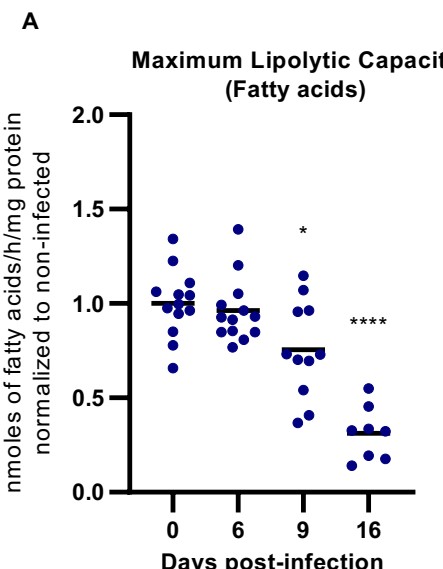

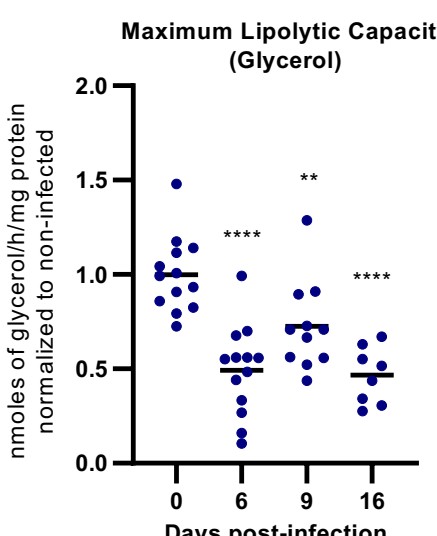

**Extended Data Fig. 4 | Effect of *T. brucei* infection on maximum lipolytic capacity.** *Ex vivo* release of **(a)** FFAs and **(b)** glycerol from AT explants under 20 μM stimulation with forskolin. Data are normalized to non-infected controls. n(day 0; 6; 9; 16) = 13, 13, 11, 8 mice examined over 2 independent experiments.).

Statistical analysis was performed with one-way ANOVA using Sidak's test for multiple comparisons. *, P < 0.05; **, P < 0.01; ****, P < 0.0001. Statistical source data contains additional parameters.

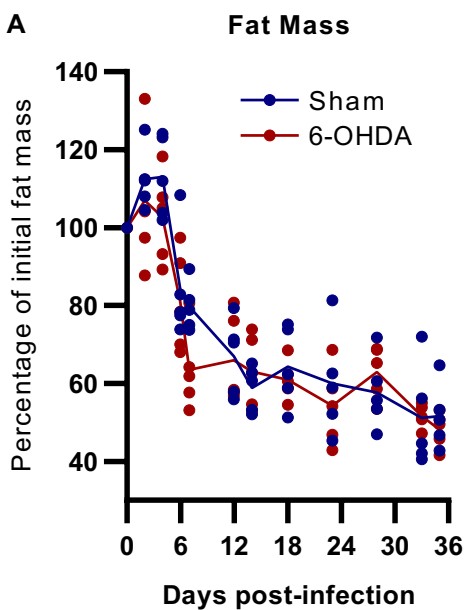

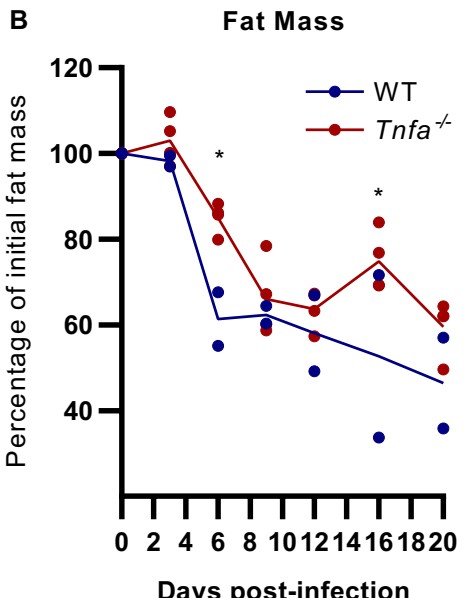

**Extended Data Fig. 5 | Effect of chemical sympathectomy and TNF-α deficiency on fat mass progression.** Fat mass relative to baseline of infected **(a)** 6-OHDA-treated or **(b)** *Tnfa⁻/⁻* mice and respective WT or sham-treated controls. **(a)** n = 5 mice per group examined in a single experiment and **(b)** n = 2 WT and 4 *Tnfa⁻/⁻* mice examined in a single experiment. Statistical analysis was performed with Two-way ANOVA using Sidak's test for multiple comparisons. *, P < 0.05. Statistical source data contains additional parameters.

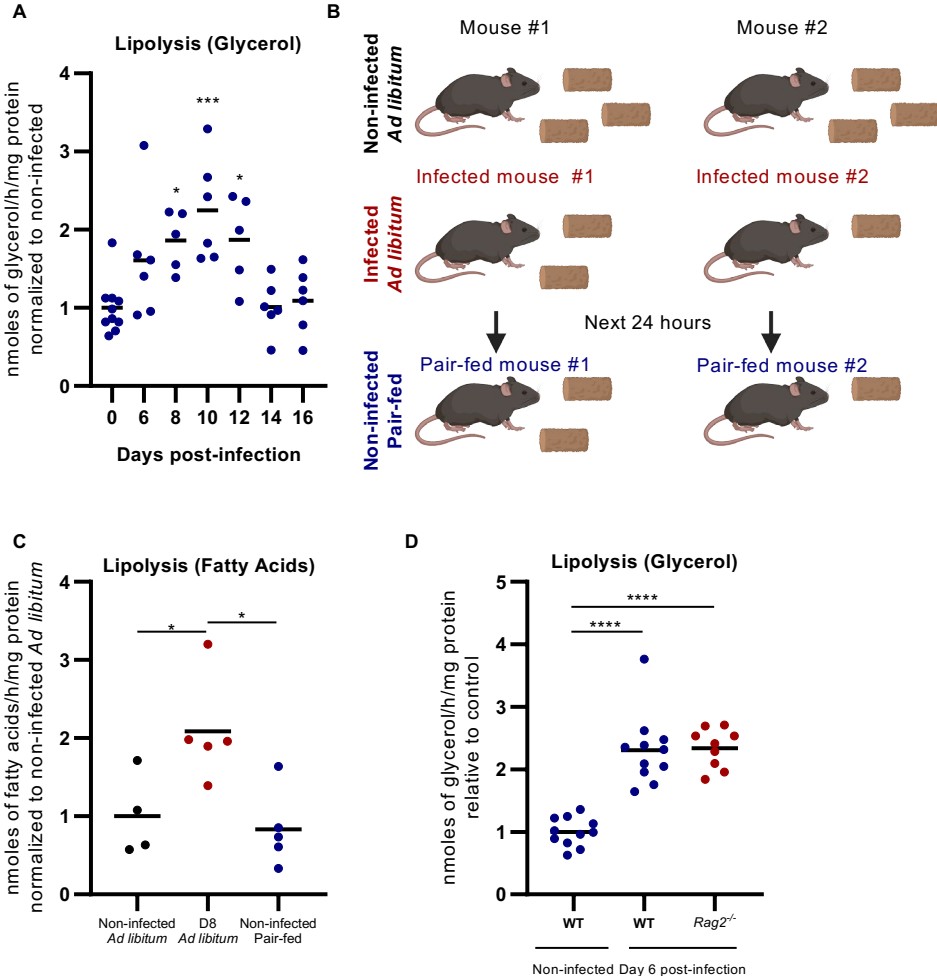

**Extended Data Fig. 6 | Impact of feeding behavior and immune modulation in *T. brucei* induced adipocyte lipolysis. (a)** Release of glycerol from gonadal AT explants of infected and non-infected WT mice under fasted and re-fed conditions, normalized to non-infected mice n(day 0;6;8;10;12;14;16) = 10, 6, 5, 6, 5, 6, 6 mice per group. **(b)** Schematic representation of the paired-feeding experimental layout. **(c)** Release of FFAs from gonadal AT explants of mice infected for 8 days and non-infected WT mice, normalized to *ad libitum* fed non-infected mice. n = 4 mice for non-infected ad libitum group and 5 mice for

infected and non-infected pair-fed groups, examined in a single experiment. **(d)** Release of glycerol from gonadal AT explants of infected WT and *Rag2⁻ᐟ⁻* mice normalized to non-infected WT controls. n = 11 non-infected WT mice, 11 infected WT mice and 9 infected Rag2-/- mice examined over 2 independent experiments. Statistical analysis was performed with **(a and c-d)** One-way ANOVA using Sidak's test for multiple comparisons. *, P < 0.05; **, P < 0.01; ***, P < 0.001. Statistical source data contains additional parameters.

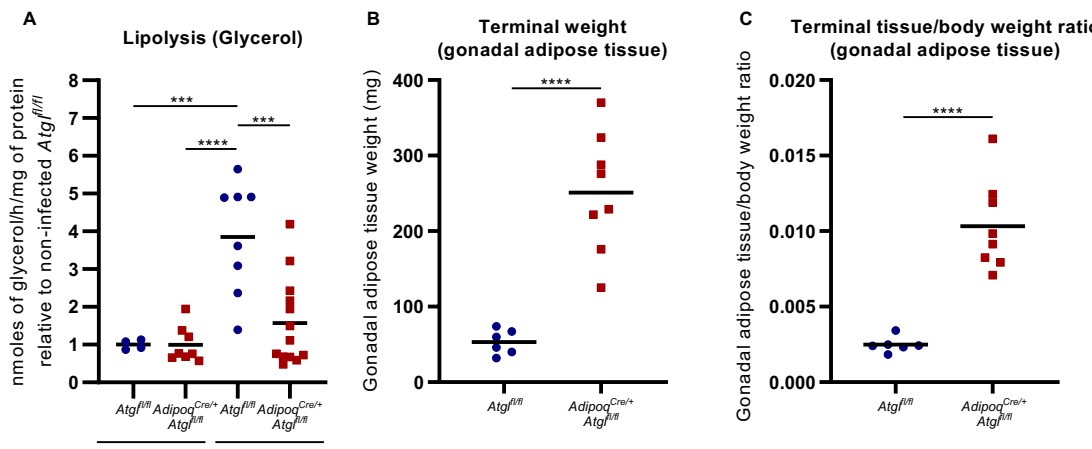

**Extended Data Fig. 7 | Effect of ATGL activity on gonadal AT glycerol release terminal weight. (a)** Release of glycerol from gonadal AT explants of infected and non-infected *Atgl*[fl/fl] (WT) and *Adipoq*[Cre/+]-*Atgl*[fl/fl] (KO) mice. n = 4 non-infected and 8 infected *Atgl*[fl/fl] mice and 8 non-infected and 13 infected *Adipoq*[Cre/+]-*Atgl*[fl/fl] mice examined over 2 independent experiments. Terminal **(b)** gonadal AT mass and **(c)** gonadal AT mass to body weight ratio of Adipoq[Cre/+]-*Atgl*[fl/fl] and *Atgl*[fl/fl]

littermate controls. n = 6 *Atgl*[fl/fl] mice and 8 *Adipoq*[Cre/+]-*Atgl*[fl/fl] examined in a single experiment. Statistical analysis was performed with **(A)** One-way ANOVA using Sidak's test for multiple comparisons and **(b-c)** two-sided unpaired t test. ***, P < 0.001; ****, P < 0.0001. Pooled data from two independent experiments. Statistical source data contains additional parameters.

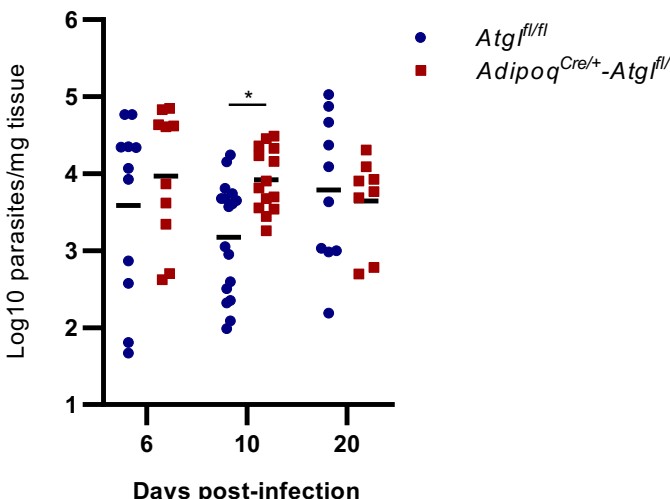

**Extended Data Fig. 8 | Effect of adipocyte-specific ATGL deficiency in AT parasite density.** Log10 number of *T. brucei* parasites per milligram of gonadal AT at different time-points post-infection of *Atgl*[fl/fl] and *Adipoq*[Cre/+]*-Atgl*[fl/fl] mice. n(day 6; 10; 20) = 11, 17, 10 *Atgl*[fl/fl] mice and 10, 14, 8 *Adipoq*[Cre/+]*-Atgl*[fl/fl] mice examined over 2 independent experiments. Statistical analysis was performed with two-way ANOVA using Sidak's test for multiple comparisons. *, $P < 0.05$. Statistical source data contains additional parameters.

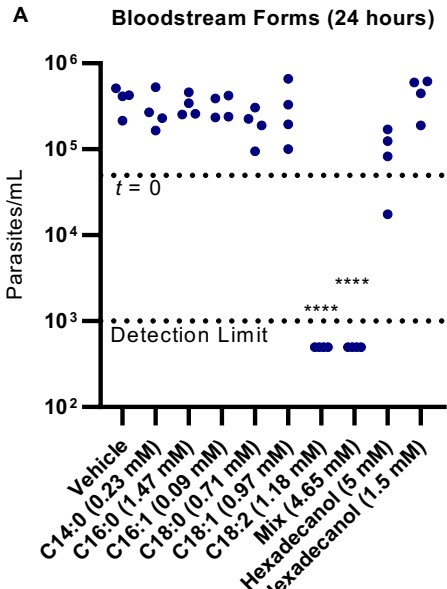
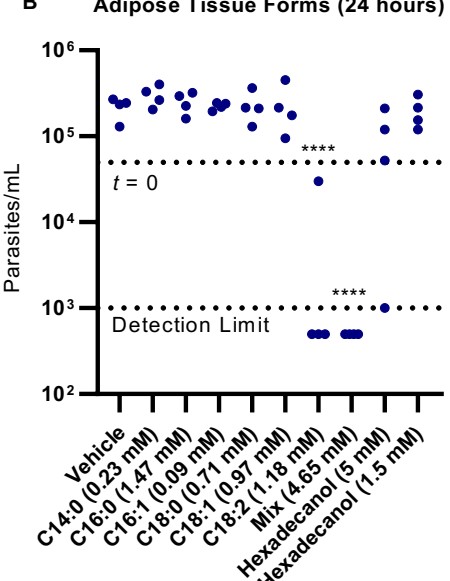

**Extended Data Fig. 9 | Effect of AT interstitial FFAs in *in vitro* parasite number.**
Number of **(a)** bloodstream form and **(b)** AT form *T. brucei* parasites after
24 hours of incubation with physiological concentrations of individual FFAs
or with an *in vivo* mimetic FFA mixture. Detection limit is 1000 parasites/mL
and initial parasite inoculum ($t = 0$) is $5 \times 10^4$ parasites/mL. Data analysed using
one-way ANOVA with Sidak's test for multiple comparisons. ****, P < 0.0001.
n = 4 independent experiments. Statistical source data contains additional
parameters.

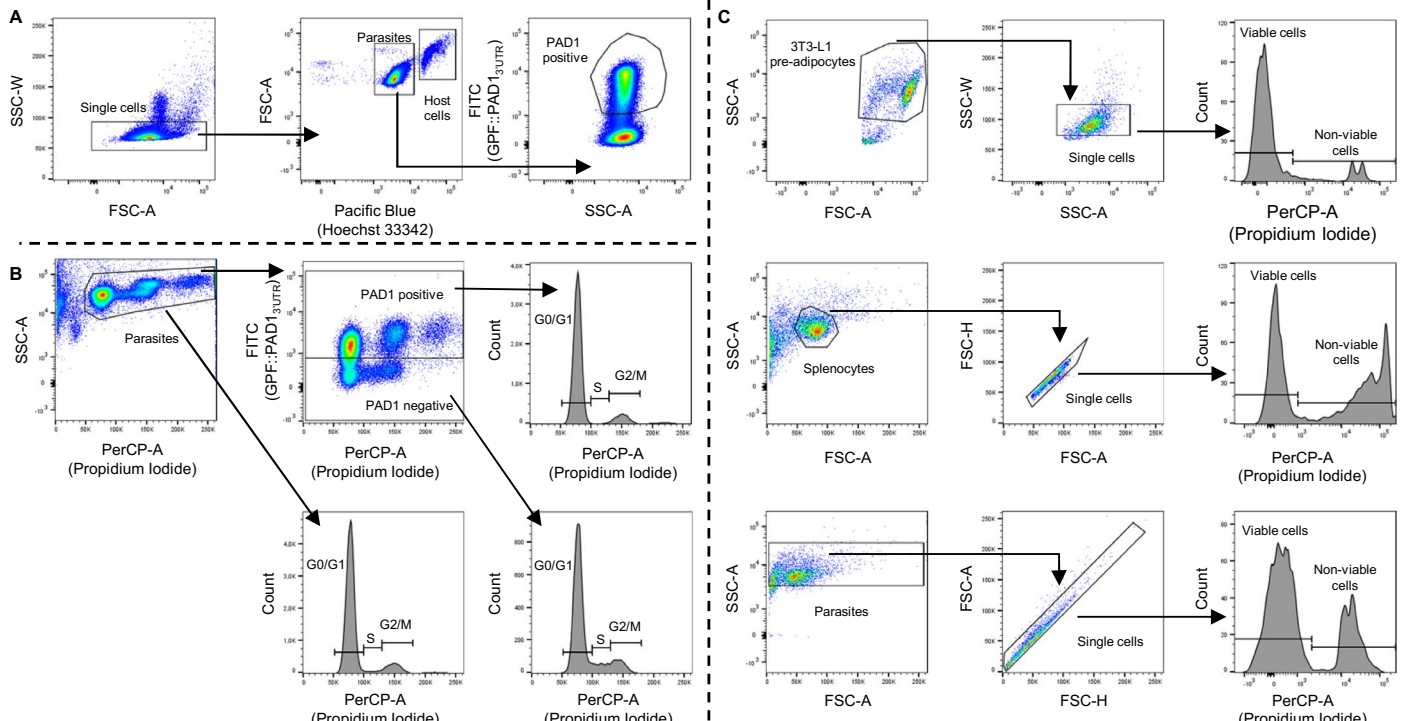

**Extended Data Fig. 10 | Flow cytometry gating strategy. (a)** Parasites and host cells isolated from the AT were differentiated based on Pacific Blue (Hoechst 33342) intensity and stumpy forms of the parasite were identified based on positive FITC (GFP::PAD1$_{3'UTR}$) signal. **(b)** Parasites were selected based on positive PerCP (propidium iodide) signal. Cell cycle was analysed in the entire parasite

population or within FITC (GFP::PAD1$_{3'UTR}$) positive and negative populations. Cell cycle stages were attributed based on PerCP (propidium iodide) signal distribution. **(c)** 3T3-L1 pre-adipocytes, splenocytes and parasites were identified based on SSC-A vs FSC-A gating. Non-viable cells were identified based on positive propidium iodide signal.

# Reporting Summary

## Statistics

For all statistical analyses, confirm that the following items are present in the figure legend, table legend, main text, or Methods section.

| n/a | Confirmed | |
|---|---|---|
| ☐ | ☒ | The exact sample size (*n*) for each experimental group/condition, given as a discrete number and unit of measurement |
| ☐ | ☒ | A statement on whether measurements were taken from distinct samples or whether the same sample was measured repeatedly |
| ☐ | ☒ | The statistical test(s) used AND whether they are one- or two-sided <br> *Only common tests should be described solely by name; describe more complex techniques in the Methods section.* |
| ☒ | ☐ | A description of all covariates tested |
| ☐ | ☒ | A description of any assumptions or corrections, such as tests of normality and adjustment for multiple comparisons |
| ☐ | ☒ | A full description of the statistical parameters including central tendency (e.g. means) or other basic estimates (e.g. regression coefficient) AND variation (e.g. standard deviation) or associated estimates of uncertainty (e.g. confidence intervals) |
| ☐ | ☒ | For null hypothesis testing, the test statistic (e.g. *F*, *t*, *r*) with confidence intervals, effect sizes, degrees of freedom and *P* value noted <br> *Give P values as exact values whenever suitable.* |
| ☒ | ☐ | For Bayesian analysis, information on the choice of priors and Markov chain Monte Carlo settings |
| ☒ | ☐ | For hierarchical and complex designs, identification of the appropriate level for tests and full reporting of outcomes |
| ☒ | ☐ | Estimates of effect sizes (e.g. Cohen's *d*, Pearson's *r*), indicating how they were calculated |

*Our web collection on statistics for biologists contains articles on many of the points above.*

## Software and code

Policy information about availability of computer code

| | |
|---|---|
| Data collection | LSRFortessa version 6.2 (BD Biosciences), NDP.scan version 1.0 (Hamamatsu Photonics); i-control version 2.0 (Tecan); Minispec Plus version 7.0.0 (Bruker); Quantstudio 5 version 1.5.1. |
| Data analysis | FlowJo v10.0.7r2 software (Tree Star); GraphPad Prism (version 8.4.3); ImageJ version 1.51h (NIH) with Adiposoft plugin (version 1.16); Quantstudio Design & Analysis software version 2.6 |

For manuscripts utilizing custom algorithms or software that are central to the research but not yet described in published literature, software must be made available to editors and reviewers. We strongly encourage code deposition in a community repository (e.g. GitHub). See the Nature Portfolio guidelines for submitting code & software for further information.

## Data

Policy information about availability of data

All manuscripts must include a data availability statement. This statement should provide the following information, where applicable:
- Accession codes, unique identifiers, or web links for publicly available datasets
- A description of any restrictions on data availability
- For clinical datasets or third party data, please ensure that the statement adheres to our policy

This study does not include large datasets. All data used in this study is available as source data files.

March 2021

## Human research participants

Policy information about studies involving human research participants and Sex and Gender in Research.

| | |
|---|---|
| Reporting on sex and gender | N/A |
| Population characteristics | N/A |
| Recruitment | N/A |
| Ethics oversight | N/A |

Note that full information on the approval of the study protocol must also be provided in the manuscript.

# Field-specific reporting

Please select the one below that is the best fit for your research. If you are not sure, read the appropriate sections before making your selection.

☒ Life sciences          ☐ Behavioural & social sciences          ☐ Ecological, evolutionary & environmental sciences

For a reference copy of the document with all sections, see nature.com/documents/nr-reporting-summary-flat.pdf

# Life sciences study design

All studies must disclose on these points even when the disclosure is negative.

| | |
|---|---|
| Sample size | Sample size was pre-estimated by a power analysis performed on G*Power 3.1 software. Most experiments were performed with at least 4 mice per group and with at least 2 independent experiments. |
| Data exclusions | Extended Data Figure 4B: Data pertaining to 1 WT mouse that unexpectedly succumbed early to infection was excluded.<br>Extended Data Figure 5A: Excluded two abnormally elevated glycerol measurements noted as outliers through Grubbs' test (1 mouse at day 8 and 1 mouse at day 12 post-infection). |
| Replication | The experimental findings were reliably reproduced as validated by at least two independent experiments. |
| Randomization | No randomization. Comparisons between infected and non-infected mice and between Atglfl/fl and AdipoqCre/+ -Atglfl/fl mice were performed using co-housed littermate controls. |
| Blinding | No blinding was performed in experiments where infected and non-infected mice were compared, as symptoms of infection allow for easy distinction between groups.<br>In experiments comparing infected Atglfl/fl and AdipoqCre/+ -Atglfl/fl mice (Fig. 3-4, 6F and Extended Data Fig.7-8), genotype information was only revealed after mice were euthanized.<br>Investigators were not blinded to group allocation for microscopy acquisitions, however downstream data processing relied on random field acquisition and automated analysis.<br>Acquisition of flow cytometry data, qPCR data and lipolysis data does not involve subjective measurements, reducing the requirement of blinding to group allocation during data collection. |

# Reporting for specific materials, systems and methods

We require information from authors about some types of materials, experimental systems and methods used in many studies. Here, indicate whether each material, system or method listed is relevant to your study. If you are not sure if a list item applies to your research, read the appropriate section before selecting a response.

## Materials & experimental systems

| n/a | Involved in the study |
|---|---|
| ☒ | ☐ Antibodies |
| ☐ | ☒ Eukaryotic cell lines |
| ☒ | ☐ Palaeontology and archaeology |
| ☐ | ☒ Animals and other organisms |
| ☒ | ☐ Clinical data |
| ☒ | ☐ Dual use research of concern |

## Methods

| n/a | Involved in the study |
|---|---|
| ☒ | ☐ ChIP-seq |
| ☐ | ☒ Flow cytometry |
| ☒ | ☐ MRI-based neuroimaging |

# Eukaryotic cell lines

Policy information about cell lines and Sex and Gender in Research

| | |
|---|---|
| Cell line source(s) | EATRO 1125 AnTat1.1E 90-13 from Keith Matthews laboratory (The University of Edinburgh, UK).<br>EATRO 1125 AnTat1.1E 90–13 GFP::PAD1 3'utr cell line from Christian Janzen laboratory (University of Wurzburg, Germany).<br>3T3-L1 (ATCC - CCL-173™. Gaithersburg, MD, USA) from Susana Constantino (University of Lisbon). |
| Authentication | The cell lines were not authenticated. |
| Mycoplasma contamination | Cell lines were not tested for mycoplasma contamination. |
| Commonly misidentified lines<br>(See ICLAC register) | No commonly misidentified lines were used in this study. |

# Animals and other research organisms

Policy information about studies involving animals; ARRIVE guidelines recommended for reporting animal research, and Sex and Gender in Research

| | |
|---|---|
| Laboratory animals | Male mice (C57BL/6J WT, Rag2-/-, Tnfa-/-) between 8–12 weeks old.<br>Sex and aged matched AdipoqCre/+ -Atglfl/fl and Atglfl/fl co-housed littermate controls were used aged between 8-20 weeks old.<br>Mice were housed in a Specific-Pathogen-Free barrier facility, under standard laboratory conditions: 21 to 22°C ambient temperature, a 12 h light/12 h dark cycle and 45 to 65% humidity. Chow and water were available ad libitum. |
| Wild animals | No wild animals were used in this study. |
| Reporting on sex | Parasite tropism towards the adipose tissue was initially described in male mice. Accordingly, whenever possible male mice were used in this work. Due to limitations in generating sufficient experimental Atglfl/fl and AdipoqCre/+-Atglfl/fl male mice, females were evenly distributed across infected and non-infected groups in figure 3, figure 4C-G, figure 6F and extended data figures 7A and 8. Sample sizes in this study are insufficient to perform post hoc analyses based on sex. |
| Field-collected samples | No field-collected samples were used in this study. |
| Ethics oversight | Animal experiments were performed according to EU regulations and approved by the Órgão Responsável pelo Bem-estar Animal (ORBEA) of Instituto de Medicina Molecular João Lobo Antunes and the competent authority Direcção Geral de Alimentação e Veterinária (licenses: 018889\2016 and 017549\2021). |

Note that full information on the approval of the study protocol must also be provided in the manuscript.

# Flow Cytometry

## Plots

Confirm that:

☒ The axis labels state the marker and fluorochrome used (e.g. CD4-FITC).

☒ The axis scales are clearly visible. Include numbers along axes only for bottom left plot of group (a 'group' is an analysis of identical markers).

☒ All plots are contour plots with outliers or pseudocolor plots.

☒ A numerical value for number of cells or percentage (with statistics) is provided.

## Methodology

| | |
|---|---|
| Sample preparation | AnTat1.1E GFP::PAD1 3'utr reporter parasites isolated from adipose tissue by gentle agitation in culture medium or from 3T3-L1 co-cultures by harvesting culture supernatants . These cells were then fixed with either formaldehyde or ethanol  and stained with Hoechst 33342 or propidium iodide and then filtered prior to acquisition.<br>AnTat1.1E parasites in axenic cultures were stained with propidium iodide and filtered prior to acquisition.<br>Splenocytes were obtained from non-infected WT mice, cell suspensions obtained through mechanical desegregation and subjected to red blood cell lysis. Splenocytes were then used in  axenic cultures, stained with propidium iodide and filtered prior to acquisition.<br>3T3-L1 pre-adipocytes were cultivated to 70-80% confluence, passaged using trypsin and used in  axenic cultures followed by staining with propidium iodide and filtered prior to acquisition. |
| Instrument | LSRFortessa (BD Biosciences) |
| Software | FACSDiva software version 6.2 (BD Biosciences) for data acquisition and FlowJo software version 10.0.7r2 (Tree Star) for analysis. |
| Cell population abundance | No cell sorting was performed in this study. |

Gating strategy

After excluding doublets through SSC-W and SSC-A gating, stumpy forms were identified as Hoescht intermediate and PAD1 positive.
Non-viable parasites were identified based on positive propidium iodide signal.
Boundaries between positive and negative populations were determined using non-stained controls.
Assignment of cell cycle stages was performed based on visual of propidium iodide histogram distribution using a linear scale.

☒ Tick this box to confirm that a figure exemplifying the gating strategy is provided in the Supplementary Information.

