## [Peer Review File · Nature Microbiology]

Peer Review Information

Journal: Nature Microbiology

Manuscript Title: Adipocyte lipolysis protects mice against Trypanosoma brucei infection

Corresponding author name(s): Luísa M. Figueiredo

Reviewer Comments & Decisions:

Decision Letter, initial version:

Message: 17th February 2023

Dear Professor Figueiredo,

Thank you for your patience while your manuscript "Adipocyte lipolysis protects the host against Trypanosoma brucei infection" was under peer-review at Nature Microbiology. It has now been seen by 4 referees, whose expertise and comments you will find at the end of this email. Although they find your work of some potential interest, they have raised a number of concerns that will need to be addressed before we can consider publication of the work in Nature Microbiology.

In particular, referee #1 is concerned that the high in vitro concentrations of free fatty acids required for parasite killing do not reflect physiological conditions in adipocyte tissue. Further this referee asks whether the trypanosomes used in the analysis of FA toxicity best mimic those found in adipose tissue. Referee #2 states that some of the in vitro experiments require additional biological replicates, and that it is not clear how error/statistical analyses have been calculated. Further, referee #2 states that western blot and QPCR experiments to evaluate the expression of lipid uptake machinery, storage machinery and phosphorylation of lipolysis enzymes seem important to substantiate the aspects of adipocyte biology that are dysregulated and whether this is limited to adipose depots. This referee also says that the data connecting the in vitro and in vivo phenotypes are weak and that it is possible that metabolites other than lipids are directly cytotoxic to T. brucei. Furthermore, referee #2 states that there is no data to support the in vivo relevance of the concentrations (0.5-1 μ M) needed to restrict T. brucei. Referee #3 suggests to discuss the possible implications of the observed increased glycerol levels. Referee #4 has some concerns regarding the overall level of conceptual advance and level of mechanistic insight provided. In further comments to the editor, the referee also mentioned some concerns regarding the physiological relevance of the high concentrations of FFA used, and the lack of a non-FFA control in the FFA experiments.

Should further experimental data allow you to address these criticisms, we would be happy to look at a revised manuscript.

Please include a data availability statement as a separate section after Methods but before references, under the heading "Data Availability". This section should inform readers about the availability of the data used to support the conclusions of your study. This information includes accession codes to public repositories (data banks for protein, DNA or RNA sequences, microarray, proteomics data etc...), references to source data published alongside the paper, unique identifiers such as URLs to data repository entries, or data set DOIs, and any other statement about data availability. At a minimum, you should include the following statement: "The data that support the findings of this study are available from the corresponding author upon request", mentioning any restrictions on availability. If DOIs are provided, we also strongly encourage including these in the Reference list (authors, title, publisher (repository name), identifier, year). For more guidance on how to write this section please see: <http://www.nature.com/authors/policies/data/data-availability-statements-data-citations.pdf>

- * If you have not done so already we suggest that you begin to revise your manuscript so that it conforms to our Article format instructions at <http://www.nature.com/nmicrobiol/info/final-submission>. Refer also to any guidelines provided in this letter.

2When submitting the revised version of your manuscript, please pay close attention to our [Digital Image Integrity Guidelines](https://www.nature.com/nature-portfolio/editorial-policies/image-integrity) and to the following points below:

Please use the link below to submit a revised paper:
[redacted]

Note: This url links to your confidential homepage and associated information about manuscripts you may have submitted or be reviewing for us. If you wish to forward this e-mail to co-authors, please delete this link to your homepage first.

Nature Microbiology is committed to improving transparency in authorship. As part of our efforts in this direction, we are now requesting that all authors identified as 'corresponding author' on published papers create and link their Open Researcher and Contributor Identifier (ORCID) with their account on the Manuscript Tracking System (MTS), prior to acceptance. This applies to primary research papers only. ORCID helps the scientific community achieve unambiguous attribution of all scholarly contributions. You can create and link your ORCID from the home page of the MTS by clicking on 'Modify my Springer Nature account'. For more information please visit www.springernature.com/orcid.

If you wish to submit a suitably revised manuscript we would hope to receive it within 6 months. If you cannot send it within this time, please let us know. We will be happy to consider your revision, even if a similar study has been accepted for publication at Nature Microbiology or published elsewhere (up to a maximum of 6 months).

Yours sincerely,
[redacted]Nature Microbiology

Reviewer Expertise:

Referee #1: Trypanosome biology

Referee #2: Innate immunity, host-parasite interactions, *Toxoplasma gondii*

Referee #3: Trypanosome metabolism

Referee #4: Trypanosome biology

Reviewer Comments:

Reviewer #1 (Remarks to the Author):

In this manuscript, Machado and co-workers build on previous work by this laboratory which led to the discovery of an adipocyte tissue form (ATF) of the African trypanosome *T. brucei* (2016). This established that these parasites could infect adipose tissue and once there could up regulate genes for fatty acid metabolism. While these studies established that the parasites actively utilized this tissue questions concerning the impact on parasite infection and host pathology remained unanswered. The current paper addresses these complex questions. Considering the wealth of data presented the paper is very clearly written and the conclusions well supported.

Several fundamental and unresolved questions are addressed in these studies. 1) Does infection of adipose tissue by the parasite contribute body wasting associated with African trypanosomiasis? The focus of the studies was on adipocyte lipolysis. Results show that early during infection that lipolysis was upregulated and dependent on host immune response. Consistent with these results disruption of the host triglyceride lipase pathway, by gene knockout, resulted in sparing infected mice with the loss of adipocyte volume. 2) Does the upregulated lipolysis in infected mice contribute to parasitemia? Ablation of adipocyte lipolysis resulted in higher parasite number in adipose tissue and earlier host death. 3) Are free fatty acids toxic to trypanosomes? At high concentrations it is proposed that free fatty acids derived from infection induced adipocyte lipolysis negatively modulates parasite numbers and is important to maintaining a chronic infection.

While I believe the results presented here are important and the studies well designed. However, I am concerned that the high invitro concentrations of free fatty acids required for parasite killing do not reflect physiological conditions in adipocyte tissue. In Figure 8 the effects on growth and PI uptake are largely restricted to the highest concentrations used (1 mM). Is this within the physiological range of free fatty acids in adipose tissue? Some discussion of the range of free FA in adipose tissue is needed.

A related question is whether the trypanosomes used in the analysis of FA toxicity best mimic those found in adipose tissue. We know, from the authors previous studies, that trypanosomes within adipose tissue adaptively change to a FA utilizing metabolism. These cells may respond very differently to free FA then bloodstream parasites tested here.

Reviewer #2 (Remarks to the Author):

This is a timely article that should be of interest to a wide range of groups studying host homeostasis and response to infection. Machado et al. show that adipose loss during *T. brucei* infection is dependent on lipolysis in adipose tissues using conditional knockouts of adipose triglyceride lipase (ATGL). Adipose loss is not simply due to anorexia, as pair-fed mice do not exhibit similar weight, and *Rag2^{-/-}* mice are protected, implicating pathological inflammatory fat loss, although the precise signaling pathway is not nailed down. These

4results are consistent with many recent reports from mouse models of endotoxemia, viral and bacterial infections are expected. The novelty of this report hinges on the unexpected observation AdipoqCr-Atg1f1/fl mice, which cannot undergo adipose depot lipolysis, have impaired survival and higher adipose parasite load compared to WT mice and other tissues. This was not simply due to a failure to convert Tb to the non-replicative form because stumpy parasites are more abundant. The authors hypothesize that fatty acids made by adipocytes are toxic to T.b. However, the link between the in vitro observation that parasites are restricted when cultured with fatty acids and the in vivo relevance of both the fatty acids selected and concentrations used is weak. In general, the paper is clearly written. In vivo experiments are adequately controlled. However, some of the in vitro experiments have not been amply repeated, and it is not clear how error/statistical analyses have been calculated and conclusions are based on single assays for lipolysis or parasite differentiation. These weaknesses do not dampen overall enthusiasm for the report if they can be addressed.

Specific Comments:

1) The authors show that blocking lipolysis in AdipoqCreAtg1f1/f mice is sufficient to worsen disease severity and increase parasite load. However, knocking out ATGL is a 'sledgehammer' to the system and the data indicating that phenotype is driven by hyper-activation of lipolysis is based on a single assay: increased NEFA and glycerol shed from explants into the supernatant. The disconnect between in vitro and in vivo glycerol levels is postulated to be due to parasite production of glycerol, though no direct evidence is provided to support this model. There are several ways that the proposed model could be tested. For example, simple western blot and QPCR experiments to evaluate the expression of lipid uptake machinery, storage machinery and phosphorylation of lipolysis enzymes seem important to substantiate the aspects of adipocyte biology that are dysregulated and whether this is limited to adipose depots.

2) The mechanistic bases for lipolysis restricting parasite growth hinges on the observation that lipids are directly inhibitory to parasite growth, however, the data connecting the in vitro and in vivo phenotypes are weak and it is possible that metabolites other than lipids are directly cytotoxic to T.b. Specifically:

-Forskolin experiments are not rigorously tested. Fig8A-C were replicated 1 or 2 times only and the application of statistics are not well described. It is possible that forskolin elicits release fundamentally distinct lipid species than the setting of infection (see below). Also, the effect of forskolin on parasite differentiation has not been adequately tested, a single measure of stumpy form differentiation (GFP) is not rigorous.

-The justification for the selection of fatty acid species or the in vivo relevance for the concentrations that inhibit parasite survival are not substantiated. The paper cited evaluated fatty acid species in the context of insulin resistance and obesity, it is very possible that these are not the same species that are abundant during parasite infection and inflammation. Moreover, even if these species are the most abundant in the adipocyte, intracellular species may not be the same as those released from the cell to interact with the extracellular parasites in vivo. There is no data to support the in vivo relevance of the concentrations (0.5-1 μ m) needed to restrict Tb. Ideally, the lipid species and concentrations would be measured directly through lipidomic approaches. However other ways of connecting the lipids released in the context of infection could also suffice, for example, transferring lipids from infected explants or rescuing parasite growth in explants

5with ATGL inhibitors.

Additional comments:

Fig. 5A and 5B show that the lack of ATGL in adipocytes results in reduced lipolysis in adipose tissue on day 6 post-infection. Fig. 8 presents that the fatty acids-induced loss of parasite viability can happen after 24 hours of incubation. Based on these data, I would expect that the reduction of parasite burden in the adipose tissue from *T. brucei*-infected AdipoqCre/+ -Atglfl/fl mice as compared to Atglfl/fl mice would start from da6 post-infection (Fig. 7C and 7D). However, the parasite reduction in AdipoqCreAtglfl/fl mice only happened on day 10 post-infection. Is there any possible mechanism leading to the delay of parasite reduction? Could you please address this in the discussion?

Page 16 2nd paragraph: The rationale for this conclusion is confusing In the AdipoqCreAtglfl/fl mice there is no lipolysis and higher parasite load so why would the 'end' of lipolysis in WT mice lead to parasite restriction-perhaps I am missing the point, please clarify in the text.

Figure 7E Plotting the fat on the same heat map for direct comparison with other tissues.

Discussion Page 25. The comparison to *Toxoplasma* is an interesting one, however, there is a compartmentalization issue in drawing this comparison because *Toxoplasma* can access intracellular host lipids but *Tb* does not, is there evidence for enhanced export in *Tb* that you discuss?

Reviewer #3 (Remarks to the Author):

This is a highly interesting, well written manuscript. In a systematic manner, with well-designed and -performed experiments involving diverse, appropriate methodologies, the authors investigated how bloodstream-form (BSF) African trypanosomes residing in adipose tissue of mice affect this tissue and how this contributes to the pathology caused by the infection. The surprising finding is that, through immune activation, adipocyte lipolysis is upregulated, causing a loss of fat mass and increased levels of free fatty acids and glycerol. The increased fatty acid levels affect parasite viability and increase the survival rate of the hosts. All necessary control experiments have been performed, and the data have been statistically analysed in an appropriate manner. The conclusions seem warranted. The topic and results are of interest of parasitologists and researchers of infectious diseases, and, thus, the manuscript is within the scope of Nature Microbiology.

It is well-known to (at least several) people studying lipid metabolism in trypanosomes that, particularly the BSFs, are susceptible to free fatty acids when added to the parasites in vitro at concentrations in the high-micromolar range. However, this is to my knowledge the first study in which this is linked to the in-situ situation and –most importantly– is shown that the host uses it to control the infection.

6I have no major comments on the manuscript. However, I would like to suggest that the authors discuss the possible implications of the observed increased glycerol levels. Fatty acids and glycerol are released from adipose triglycerides by the activation of lipolysis during infection. However, the relative increase in glycerol level is significantly higher than that of the fatty acids. This is explained by the production of glycerol by trypanosomes that is well-known to occur from glucose catabolism under hypoxic conditions. Indeed, the authors show glycerol production in cultured BSF trypanosomes in Supplementary Figure 2. My specific questions are:

1. Usually, cultured BSFs produce no or only very little glycerol, unless the oxygen concentration is very low (Ref. 18) or the oxidase is inhibited (e.g. PMID: 22580731; PMID: 24367711; PMID: 25775470). Was the oxygen supply limited in the experiment of Fig. S2? If so, was that only due to the high density of the culture? Is there information about the oxygen availability in adipose tissue?
2. In recent years, several papers have been published indicating that BSFs can grow with glycerol as alternative source for its ATP production instead of using glucose. These studies were performed in vitro with high glycerol concentrations, but it has hypothesized that it may also occur at in vivo, for example in adipose tissue. However, use and production of glycerol are incompatible; the same enzyme, glycerol kinase, is used, acting in the reverse direction for glycerol production because of the absence of a glycerol-3-phosphatase. Thus, could the authors address this issue? Do the authors consider glycerol as an unlikely energy substrate for adipose trypanosomes? Furthermore, is it possible that the accumulation of glycerol also affects the growth rate of the trypanosomes? The glycerol kinase reaction involves a large free energy change, with the reversal only possible at low (glycosomal) ATP/ADP ratio and favoured by a high G3P/glycerol ratio (PMID: 2999127).

Some minor comments:

1. Page 8, 3 lines from bottom: No panel G
2. Page 19, 3 lines from bottom and page 20 2nd line: trypanocidal should be trypanocidal.

Reviewer #4 (Remarks to the Author):

This is a follow up to earlier work demonstrating the presence of an adipose tissue reservoir and form of the African trypanosome. The possible connection with cachexia and alterations in metabolism are of obvious importance to study of pathogenesis. The work is carried out in a mouse model and exploits a couple of knockout strains.

Is this mechanistic - unclear what pathways we have or how this actually impacts the parasite - as is this is phenomenology to me. I'm also not sure what I have really learned here that extends our understanding in a manner that provides any sort of paradigm shift.

Some minor points

The pathology of African trypanosomiasis is extensively modulated by the extravascular location of the parasite.

7

Odd sentence

associated with the parasite's
Do not need the possessive here

F1 Scale bar is???
during a T. brucei
Do not need 'a'
identify whether loss
Suggests a comparison - if possibility a or b.
How FA affected but glycerol not. Explanation is possible but this needs evidence to
validate otherwise these data seem to me difficult to reconcile.
PAD 1 - reporter, not stumpy please.

Author Rebuttal to Initial comments

Reviewer 1

In this manuscript, Machado and co-workers build on previous work by this laboratory which led to the discovery of an adipocyte tissue form (ATF) of the African trypanosome *T. brucei* (2016). This established that these parasites could infect adipose tissue and once there could up regulate genes for fatty acid metabolism. While these studies established that the parasites actively utilized this tissue questions concerning the impact on parasite infection and host pathology remained unanswered. The current paper addresses these complex questions. Considering the wealth of data presented the paper is very clearly written and the conclusions well supported.

Several fundamental and unresolved questions are addressed in these studies. 1) Does infection of adipose tissue by the parasite contribute body wasting associated with African trypanosomiasis? The focus of the studies was on adipocyte lipolysis. Results show that early during infection that lipolysis was upregulated and dependent on host immune response. Consistent with these results disruption of the host triglyceride lipase pathway, by gene knockout, resulted in sparing infected mice with the loss of adipocyte volume. 2) Does the upregulated lipolysis in infected mice contribute to parasitemia? Ablation of adipocyte lipolysis resulted in higher parasite number in adipose tissue and earlier host death. 3) Are free fatty acids toxic to trypanosomes? At high concentrations it is proposed that free fatty acids derived

8from infection induced adipocyte lipolysis negatively modulates parasite numbers and is important to maintaining a chronic infection.

1. While I believe the results presented here are important and the studies well designed. However, I am concerned that the high *in vitro* concentrations of free fatty acids required for parasite killing do not reflect physiological conditions in adipocyte tissue. In Figure 8 the effects on growth and PI uptake are largely restricted to the highest concentrations used (1 mM). Is this within the physiological range of free fatty acids in adipose tissue? Some discussion of the range of free FA in adipose tissue is needed.

This important point was raised by several reviewers. To the best of our knowledge, the composition and concentration of free fatty acid in extracellular fluid of adipose tissue had never been measured in any context. Most studies characterize the whole tissue lipid environment, which is likely mostly a reflection of the adipocyte intracellular lipid composition. Thus, we had to develop a new methodology to detect and quantify the free the most abundant free fatty acid in the interstitial fluid of adipose tissue. We adapted a recently published method (MiQenbuhler et al. 2023; DOI: 10.1016/j.cmet.2022.12.014), which allows the recovery of ~1 μ L of extracellular fluid from one gonadal AT. Next, in collaboration with Terry Smith, we performed GC-MS to establish which fatty acids species are present in the interstitial fluid and at what concentration, utilizing a non-natural internal standard. This data is presented in the new panels of Figure 6. Essentially, we found that the most abundant FFA are C16:0 > C18:2 > C18:1 > C18:0 > C14:0 > C16:1 and their concentration range from 1.8 mM for C16:0 to 0.2 mM for C16:1 in non-infected mice.

Next, we repeated the toxicity assay *in vitro* using the physiological concentrations of these six species of FFA (Figure 6). We also used a novel batch of BSA to ensure that most FFA is correctly bound to carrier protein, reducing possible unspecific “detergent” effects promoted by mycelia (as suggested by reviewer 4). In these new conditions, we found that the toxicity of FFA is not as widespread as we had initially observed. Instead, we found that most FFA had no toxic effect on parasites, except for C18:2, the second most abundant FFA species in the extracellular fluid of adipose tissue (1.18 mM). The fact that physiological concentrations of C16:0 (1.47 mM) do not have a toxic effect, shows that FFA toxicity is not a mere consequence of high concentration of fatty acid, but it is rather fatty acid species specific.

2. A related question is whether the trypanosomes used in the analysis of FA toxicity best mimic those found in adipose tissue. We know, from the authors previous studies, that trypanosomes within adipose tissue adaptively change to a FA utilizing metabolism. These cells may respond very differently to free FA then bloodstream parasites tested here.

To test if parasites isolated from blood and adipose tissue are equally sensitive to FFA toxic effects, we performed the in vitro toxicity assay with either bloodstream forms (BSF) or adipose tissue forms (ATF). We found that parasites isolated from both tissues are equally sensitive to C18:2 and equally resistant to each of the five other FFA species tested in this study (Figure 6). Together with our previous results published in Trindade et al, 2016, it appears that when parasites colonize adipose tissue, their biochemical capacity of catabolizing C14:0 (myristate) is increased, but not their resistance to toxic extracellular FFA. This point is now included in the Discussion (lines 522-536).

Reviewer 2

This is a timely article that should be of interest to a wide range of groups studying host homeostasis and response to infection. Machado et al. show that adipose loss during *T. brucei* infection is dependent on lipolysis in adipose tissues using conditional knockouts of adipose triglyceride lipase (ATGL). Adipose loss is not simply due to anorexia, as pair-fed mice do not exhibit similar weight, and *Rag2*^{-/-} mice are protected, implicating pathological inflammatory fat loss, although the precise signaling pathway is not nailed down. These results are consistent with many recent reports from mouse models of endotoxemia, viral and bacterial infections are expected. The novelty of this report hinges on the unexpected observation *Adipoq*^{Cr-Atg1f1/fl} mice, which cannot undergo adipose depot lipolysis, have impaired survival and higher adipose parasite load compared to WT mice and other tissues. This was not simply due to a failure to convert Tb to the non-replicative form because stumpy parasites are more abundant. The authors hypothesize that fatty acids made by adipocytes are toxic to T.b. However, the link between the in vitro observation that parasites are restricted when cultured with fatty acids and the in vivo relevance of both the fatty acids selected and concentrations used is weak. In general, the paper is clearly written. In vivo experiments are adequately controlled. However, some of the in vitro experiments have not been amply repeated, and it is not clear how error/statistical analyses have been calculated and conclusions are based on single assays for lipolysis or parasite differentiation. These weaknesses do not dampen overall enthusiasm for the report if they can be addressed.

Specific Comments:

1) The authors show that blocking lipolysis in *Adipoq*^{Cr-Atg1f1/f} mice is sufficient to worsen disease severity and increase parasite load. However, knocking out ATGL is a 'sledgehammer' to the system and the data indicating that phenotype is driven by hyper-activation of lipolysis is

10

based on a single assay: increased NEFA and glycerol shed from explants into the supernatant. The disconnect between *in vitro* and *in vivo* glycerol levels is postulated to be due to parasite production of glycerol, though no direct evidence is provided to support this model. There are several ways that the proposed model could be tested. For example, simple western blot and QPCR experiments to evaluate the expression of lipid uptake machinery, storage machinery and phosphorylation of lipolysis enzymes seem important to substantiate the aspects of adipocyte biology that are dysregulated and whether this is limited to adipose depots.

As mentioned by the cited refs. 38 and 39 and by reviewer #3, it has been previously shown that *T. brucei* parasites can produce glycerol as a by-product of glycolysis, especially under low oxygen concentrations. Nevertheless, we decided to test this directly with our parasite strain. EATRO1125 AnTat1.1E bloodstream form parasites were cultured axenically in HMI11, in the presence of 5% or 21% oxygen for 24 hr (new Supplementary Fig. 2B). In both conditions, glycerol concentration increased showing that parasites can secrete Glycerol *in vitro* and this effect is more pronounced under low oxygen conditions. These results were reproduced with MAK65 and MAK98, two independent *T. brucei* strains recently isolated from cows in Uganda (data shown below, but not included in the paper because we only have $n=1$). Overall, Supplementary Fig. 2B provides direct evidence that Trypanosomes can secrete Glycerol, thus justifying the discrepancy between FFA and Glycerol measurements.

While our data demonstrates that blocking ATGL-mediated lipolysis is sufficient to worsen disease severity and increase parasite load in tissue, we agree with the reviewer that we cannot exclude that other pathways may also contribute to this phenotype. Measuring the secretion of NEFA and glycerol is the standard assay to measure lipolysis activity (DOI: 10.1016/B978-0-12-

800280-3.00010-4). The experiments suggested by the reviewer (qPCR and Western blotting) would inform which other lipid-related pathways may be affected, but would not address their importance (KO mice would be required). Moreover, qPCR and Western blotting have been recently undertaken by Redford SE *et al* (bioRxiv, DOI: 10.1101/2022.12.17.520896) in whole tissue samples. qPCR showed a reduced expression of genes involved in lipogenesis and no effect on lipolysis. Western blotting showed an activation of lipolysis. While there is a consistency of phenotypes in the two studies, our data does not exclude that other lipid-based pathways may also contribute to the fat wasting. This point has now been included in the Discussion (lines 478-494).

2) The mechanistic bases for lipolysis restricting parasite growth hinges on the observation that lipids are directly inhibitory to parasite growth, however, the data connecting the in vitro and in vivo phenotypes are weak and it is possible that metabolites other than lipids are directly cytotoxic to T.b. Specifically:

-Forskolin experiments are not rigorously tested. Fig8A-C were replicated 1 or 2 times only and the application of statistics are not well described. It is possible that forskolin elicits release fundamentally distinct lipid species than the setting of infection (see below). Also, the effect of forskolin on parasite differentiation has not been adequately tested, a single measure of stumpy form differentiation (GFP) is not rigorous.

Forskolin experiments (Figure 5) have now been repeated 2-4 times. Statistical analysis is described in legend of Figure 5. Quantification of slender form and stumpy forms is now based not only in GFP::PAD1_{3UTR} expression, but also on cell cycle analysis (new panel, Figure 5E-F).

-The justification for the selection of fatty acid species or the in vivo relevance for the concentrations that inhibit parasite survival are not substantiated. The paper cited evaluated fatty acid species in the context of insulin resistance and obesity, it is very possible that these are not the same species that are abundant during parasite infection and inflammation. Moreover, even if these species are the most abundant in the adipocyte, intracellular species may not be the same as those released from the cell to interact with the extracellular parasites in vivo. There is no data to support the in vivo relevance of the concentrations (0.5-1 μ m) needed to restrict Tb. Ideally, the lipid species and concentrations would be measured directly through lipidomic approaches.

This important point was also raised by reviewer 1. We repeat below our response, which describes the experimental approach and results.

This important point was raised by several reviewers. To the best of our knowledge, the composition and concentration of free fatty acid in extracellular fluid of adipose tissue had never been measured in any context. Most studies characterize the whole tissue lipid environment, which is likely mostly a reflection of the adipocyte intracellular lipid composition. Thus, we had to develop a new methodology to detect and quantify the free the most abundant free fatty acid in the interstitial fluid of adipose tissue. We adapted a recently published method (MiQenbuhler et al. 2023; DOI: 10.1016/j.cmet.2022.12.014), which allows the recovery of ~1 μ L of extracellular fluid from one gonadal AT. Next, in collaboration with Terry Smith, we performed GC-MS to establish which fatty acids species are present in the interstitial fluid and at what concentration, utilizing a non-natural internal standard. This data is presented in the new panels of Figure 6. Essentially, we found that the most abundant FFA are C16:0 > C18:2 > C18:1 > C18:0 > C14:0 > C16:1 and their concentration range from 1.8 mM for C16:0 to 0.2 mM for C16:1 in non-infected mice.

Next, we repeated the toxicity assay *in vitro* using the physiological concentrations of these six species of FFA (Figure 6). We also used a novel batch of BSA to ensure that most FFA is correctly bound to carrier protein, reducing possible unspecific “detergent” effects promoted by mycelia (as suggested by reviewer 4). In these new conditions, we found that the toxicity of FFA is not as widespread as we had initially observed. Instead, we found that most FFA had no toxic effect on parasites, except for C18:2, the second most abundant FFA species in the extracellular fluid of adipose tissue (1.18mM). The fact that physiological concentrations of C16:0 (1.47mM) do not have a toxic effect, shows that FFA toxicity is not a mere consequence of high concentration of lipids, but it is rather species specific.

However other ways of connecting the lipids released in the context of infection could also suffice, for example, transferring lipids from infected explants or rescuing parasite growth in explants with ATGL inhibitors.

We attempted to rescue parasite growth defect under co-culture conditions with 3T3-L1 adipocytes or adipose tissue explants. However, the ATGL inhibitor (Atglistatin) and the HSL inhibitor (sc-206328) were toxic for the parasite, which prevent any meaningful interpretation of the data.

While transferring lipids from infected explants to a culture setting would be an elegant solution to link the *in vitro* and *in vivo* phenotypes, the limitations of interstitial fluid we are able to harvest from mice fat pads is incompatible with an *in vitro* culture containing a meaningful ratio of culture media to interstitial fluid.

Fig. 5A and 5B show that the lack of ATGL in adipocytes results in reduced lipolysis in adipose tissue on day 6 post-infection. Fig. 8 presents that the fatty acids-induced loss of parasite viability can happen after 24 hours of incubation. Based on these data, I would expect that the reduction of parasite burden in the adipose tissue from *T. brucei*-infected AdipoqCre/+Atg1f1/fl mice as compared to Atg1f1/fl mice would start from day 6 post-infection (Fig. 7C and 7D). However, the parasite reduction in AdipoqCreAtg1f1/fl mice only happened on day 10 post-infection. Is there any possible mechanism leading to the delay of parasite reduction? Could you please address this in the discussion?

Based on new data obtained during revision, we have now shown that even basal levels of lipolysis (on day 0) are sufficient to create a toxic environment to the parasites. Despite linoleic acid being toxic to the parasites, between 10^5 - 10^6 parasites are still detectable in the adipose tissue of wild-type animals (Fig. 4C), indicating that either there is a subpopulation of parasites that resist killing and/or that there is a constant influx of parasites from the blood that replenishes the adipose tissue. The influx of parasites is probably predominant early in infection because parasitemia is also growing exponentially in the blood, peaking between days 5 and 6. Thus, the net effect of lipid toxicity might be relatively minor during those first 6 days of infection. This point has been added to the Discussion (lines 537-551).

Other technical parameters can contribute to slight time-differences between our *in vitro* and *in vivo* data. We quantify parasite load by qPCR of parasite genomic DNA and thus, if DNA is not quickly eliminated, we may overestimate parasite load. Also we cannot exclude the possibility that the time it takes to kill parasites *in vivo* is different from *in vitro*, especially as the fatty acid carrier used in our *in vitro* experiments (i.e. albumin) may be distinct from those present in the adipose tissue. Finally, *in vivo*, there may be a very rapid flux of FFA (via ceramides, lipoproteins, etc) that we do not detect with a static measurement.

Page 16 2nd paragraph: The rationale for this conclusion is confusing. In the AdipoqCreAtg1f1/fl mice there is no lipolysis and higher parasite load so why would the 'end' of lipolysis in WT mice lead to parasite restriction-perhaps I am missing the point, please clarify in the text.

Later in infection (after day 14), when adipocytes have exhausted most of their TAG storage and lipolysis is no longer detected, no lipid toxicity is expected and other mechanisms of parasite control probably become more predominant. Indeed, we have previously shown that the number of immune cells in adipose tissue is significantly higher later in infection than on day

6. Thus, later in infection, parasite load probably remains controlled exclusively by the immune response. This point has been clarified in the Discussion (lines 551-557).

Figure 7E Plotting the fat on the same heat map for direct comparison with other tissues. Given the instructions of the journal, data has been replotted displaying with individual data points.

Discussion Page 25. The comparison to Toxoplasma is an interesting one, however, there is a compartmentalization issue in drawing this comparison because Toxoplasma can access intracellular host lipids but Tb does not, is there evidence for enhanced export in Tb that you discuss?

The mass spectrometry analysis of free fatty acids present in the interstitial fluid of adipose tissue shows that on day 6 there is sufficient C18:2 free fatty acid to be toxic for the *T. brucei* parasite. Thus, it appears that the host can interfere with parasite growth both in intracellular (Toxoplasma) and extracellular (*T. brucei*) compartments.

Reviewer 3

This is a highly interesting, well written manuscript. In a systematic manner, with well-designed and -performed experiments involving diverse, appropriate methodologies, the authors investigated how bloodstream-form (BSF) African trypanosomes residing in adipose tissue of mice affect this tissue and how this contributes to the pathology caused by the infection. The surprising finding is that, through immune activation, adipocyte lipolysis is upregulated, causing a loss of fat mass and increased levels of free fatty acids and glycerol. The increased fatty acid levels affect parasite viability and increase the survival rate of the hosts. All necessary control experiments have been performed, and the data have been statistically analysed in an appropriate manner. The conclusions seem warranted.

The topic and results are of interest of parasitologists and researchers of infectious diseases, and, thus, the manuscript is within the scope of Nature Microbiology.

It is well-known to (at least several) people studying lipid metabolism in trypanosomes that, particularly the BSFs, are susceptible to free fatty acids when added to the parasites in vitro at concentrations in the high-micromolar range. However, this is to my knowledge the first study in which this is linked to the in-situ situation and –most importantly–is shown that the host uses it

15to control the infection.

I have no major comments on the manuscript. However, I would like to suggest that the authors discuss the possible implications of the observed increased glycerol levels. Fatty acids and glycerol are released from adipose triglycerides by the activation of lipolysis during infection. However, the relative increase in glycerol level is significantly higher than that of the fatty acids. This is explained by the production of glycerol by trypanosomes that is well-known to occur from glucose catabolism under hypoxic conditions. Indeed, the authors show glycerol production in cultured BSF trypanosomes in Supplementary Figure 2. My specific questions are:

1. Usually, cultured BSFs produce no or only very little glycerol, unless the oxygen concentration is very low (Ref. 18) or the oxidase is inhibited (e.g. PMID: 22580731; PMID: 24367711; PMID: 25775470). Was the oxygen supply limited in the experiment of Fig. S2? If so, was that only due to the high density of the culture?

Riley et al, 1956 showed that *T. brucei* BSF catabolize most glucose into pyruvate and glycerol. In the presence of oxygen, the ratio Pyruvate/Glycerol is 1,43/0,58 per mol of Glucose used. While in the absence of oxygen, the ratio is 0,87/1,13. While it is true that Glycerol production is enhanced in the absence of oxygen, a non-negligible amount of Glycerol is also produced in aerobic conditions. Similar results were observed by Grant et al, 1956 for *T. rhodesiense* (Table 1). Our results are consistent with this literature.

Nevertheless, we performed an additional experiment to test whether the EATRO1125 parasite strain is also capable of secreting glycerol *in vitro*. As already described above in response to reviewer #1, EATRO1125 AnTat1.1E bloodstream form parasites were cultured axenically in HMI11, in the presence of 5% or 21% oxygen for 24 hr (new Supplementary Figure 2B). These results were reproduced with MAK65 and MAK98, two independent *T. brucei* strains recently isolated from cows in Uganda (data shown above, but not included in the paper because we only have n=1). In both conditions, glycerol concentration increased, showing that parasites can secrete Glycerol *in vitro*. Importantly, glycerol secretion was more pronounced when concentration of oxygen was lower, consistent with literature. This point has been included in the Discussion (lines 558-568).

Is there information about the oxygen availability in adipose tissue?

In the literature, oxygen pressure has been measured to compare the hypoxia conditions in wild-type and ob/ob animals. Wildtype animals show 6.4-7.7% Oxygen in adipose tissue and ~2.8% in veins. Levels in the aorta normally range around 12.5-15% (DOI: 10.1152/ajpendo.90760.2008; 10.1152/ajpendo.00435.2007). While oxygen availability in the AT's interstitial spaces during *T. brucei* infection has yet to be measured, our results of Supplementary Fig. 2 suggest that parasites are likely to secrete glycerol not only in adipose tissue, but also in the vasculature. Future studies will need to consider the fate of such glycerol, which could be taken by the liver and converted to glucose through gluconeogenesis.

2. In recent years, several papers have been published indicating that BSFs can grow with glycerol as alternative source for its ATP production instead of using glucose. These studies were performed in vitro with high glycerol concentrations, but it has hypothesized that it may also occur at in vivo, for example in adipose tissue. However, use and production of glycerol are incompatible; the same enzyme, glycerol kinase, is used, acting in the reverse direction for glycerol production because of the absence of a glycerol-3-phosphatase. Thus, could the authors address this issue? Do the authors consider glycerol as an unlikely energy substrate for adipose trypanosomes? **Furthermore, is it possible that the accumulation of glycerol also affects the growth rate of the trypanosomes?** The glycerol kinase reaction involves a large free energy change, with the reversal only possible at low (glycosomal) ATP/ADP ratio and favoured by a high G3P/glycerol ratio (PMID: 2999127).

In this work, we focused on the role of fatty acids. Future studies will be necessary to dissect the fascinating role of Glycerol. From this study, it is clear that *T. brucei* secretes glycerol in the

17presence of 21% of oxygen, but even more if oxygen concentration is reduced to 5%. Thus, it is likely that within tissues, glycerol secretion is enhanced. Unpublished data from our lab has also shown very high concentrations of glycerol (10mM) can trigger differentiation of parasites, suggesting that this metabolite may have a role beyond being an additional source of carbon. In collaboration with another lab, we are currently evaluating the *in vivo* relevance of such phenotype. This point has been included in the Discussion (lines 558-568).

Some minor comments:

1. Page 8, 3 lines from bottom: No panel G

This error has been corrected.

2. Page 19, 3 lines from bottom and page 20 2nd line: trypanocidal should be trypanocidal.

This error has been corrected.

Reviewer 4

This is a follow up to earlier work demonstrating the presence of an adipose tissue reservoir and form of the African trypanosome. The possible connection with cachexia and alterations in metabolism are of obvious importance to study of pathogenesis. The work is carried out in a mouse model and exploits a couple of knockout strains.

Is this mechanistic - unclear what pathways we have or how this actually impacts the parasite - as is this is phenomenology to me. I'm also not sure what I have really learned here that extends our understanding in a manner that provides any sort of paradigm shift.

Some minor points:

The pathology of African trypanosomiasis is extensively modulated by the extravascular location of the parasite. Odd sentence.

This sentence has been removed.

“associated with the parasite’s”. Do not need the possessive here

This error has been corrected.

F1 Scale bar is???

This information has now been provided.

“during a *T. brucei*”. Do not need ‘a’

This error has been corrected.

“identify whether loss” Suggests a comparison - if possibility a or b.

This sentence has been corrected.

How FA affected but glycerol not. Explanation is possible but this needs evidence to validate otherwise these data seem to me difficult to reconcile.

We have performed additional experiments to show that *T. brucei* secrete glycerol, in an oxygen dependent manner (Supplementary Figure 2).

PAD 1 - reporter, not stumpy please.

The characterization of stumpy forms has been extended to include not only expression of PAD1::GFP reporter, but also cell cycle information (Figure 5E-F).

The editor mentioned the need to have a non-FFA control in the FFA experiments. To distinguish whether the effect of released FFA is either because they act as a “detergent” or because FFA are imported by the parasite and have a cytotoxic effect intracellularly, we measured if parasite growth is affected in vitro by hexadecanol, a non-FFA molecule that is chemically similar to a FFA (given its size and hydrophobicity) (new Figure 6). This experiment shows that, while 1.18 mM of C18:2 is toxic to the parasites, 1.5 mM hexadecanol is not. Besides, in the revised data of Figure 6, we show that only C18:2 is toxic for parasite growth. None of the other tested FFA are toxic at physiological concentrations, further showing that C18:2 toxicity is specific and thus unlikely to act as a “detergent”.

Decision Letter, first revision:

Message: Our ref: NMICROBIOL-22123122A

22nd August 2023

Dear Luisa,

Thank you for your patience as we've prepared the guidelines for final submission of your Nature Microbiology manuscript, "Adipocyte lipolysis protects the host against *Trypanosoma brucei* infection" (NMICROBIOL-22123122A). Please carefully follow the step-by-step instructions provided in the attached file, and add a response in each row of the table to indicate the changes that you have made. Ensuring that each point is addressed will help to ensure that your revised manuscript can be swiftly handed over to our production team.

In recognition of the time and expertise our reviewers provide to Nature Microbiology's editorial process, we would like to formally acknowledge their contribution to the external peer review of your manuscript entitled "Adipocyte lipolysis protects the host against *Trypanosoma brucei* infection". For those reviewers who give their assent, we will be publishing their names alongside the published article.

Nature Microbiology offers a Transparent Peer Review option for new original research manuscripts submitted after December 1st, 2019. As part of this initiative, we encourage our authors to support increased transparency into the peer review process by agreeing to have the reviewer comments, author rebuttal letters, and editorial decision letters published as a Supplementary item. When you submit your final files please clearly state in your cover letter whether or not you would like to participate in this initiative. Please note that failure to state your preference will result in delays in accepting your manuscript for publication.

Cover suggestions

COVER ARTWORK: We welcome submissions of artwork for consideration for our cover. For

20more information, please see our [guide for cover artwork](https://www.nature.com/documents/Nature_covers_author_guide.pdf).

Nature Microbiology has now transitioned to a unified Rights Collection system which will allow our Author Services team to quickly and easily collect the rights and permissions required to publish your work. Approximately 10 days after your paper is formally accepted, you will receive an email in providing you with a link to complete the grant of rights. If your paper is eligible for Open Access, our Author Services team will also be in touch regarding any additional information that may be required to arrange payment for your article.

Please note that *Nature Microbiology* is a Transformative Journal (TJ). Authors may publish their research with us through the traditional subscription access route or make their paper immediately open access through payment of an article-processing charge (APC). Authors will not be required to make a final decision about access to their article until it has been accepted. [Find out more about Transformative Journals](https://www.springernature.com/gp/open-research/transformative-journals)

Authors may need to take specific actions to achieve [compliance with funder and institutional open access mandates](https://www.springernature.com/gp/open-research/funding/policy-compliance-faqs). If your research is supported by a funder that requires immediate open access (e.g. according to [Plan S principles](https://www.springernature.com/gp/open-research/plan-s-compliance)) then you should select the gold OA route, and we will direct you to the compliant route where possible. For authors selecting the subscription publication route, the journal's standard licensing terms will need to be accepted, including [self-archiving policies](https://www.nature.com/nature-portfolio/editorial-policies/self-archiving-and-license-to-publish). Those licensing terms will supersede any other terms that the author or any third party may assert apply to any version of the manuscript.

[redacted]

Best regards,

21[redacted]

Reviewer #2:

Remarks to the Author:

This revision represents a substantially improved version of an interesting story that adequately addresses the reviewers' concerns. In the resubmission, they developed a new methodology to measure lipid species in interstitial fluid and shown that one of the differentially enriched lipids C18:2 is sufficient to impair parasite growth at levels comparable to those observed in vivo. This is a substantial improvement to the mechanistic basis of the manuscript that better substantiates a major conclusion and will be of high interest to the readers of Nature Microbiology. In addition, the authors experimentally resolved a discrepancy in the glycerol levels, demonstrating that several isolate release glycerol albeit at levels substantially lower than those that induce differentiation. The rationale for the focus on fatty acids in the discussion and other requested clarifications has been met.

Reviewer #3:

Remarks to the Author:

The authors have dealt appropriately with my comments on the originally submitted manuscript. The modifications made in response to my previous comments and those by the reviewers have improved the manuscript and strengthened the conclusions drawn.

I just spotted a few very minor issues:

1. Page 8 and Fig. 2: No reference to Figures 2B and 2C is made in the text.
2. Line 428: trypanocidal should be trypanocidal.
3. Line 512: Fig. 9H? This figure does not exist. Should it be 6F?
4. Line 801: Add 'glycerol' after (B).
5. Lines 820 and 824: The legend mentioned 'release of fatty acids', but Supplementary Figures 5A and 5D show release (or rather presence) of glycerol.
6. Line 834: 'fatty acids' should be replaced by 'glycerol' (shown in Supplementary Fig. 6).

Reviewer #4:

None

Author Rebuttal, first revision:

Reviewer #2:

Remarks to the Author:

This revision represents a substantially improved version of an interesting story that adequately addresses

22the reviewers' concerns. In the resubmission, they developed a new methodology to measure lipid species in interstitial fluid and shown that one of the differentially enriched lipids C18:2 is sufficient to impair parasite growth at levels comparable to those observed in vivo. This is a substantial improvement to the mechanistic basis of the manuscript that better substantiates a major conclusion and will be of high interest to the readers of Nature Microbiology. In addition, the authors experimentally resolved a discrepancy in the glycerol levels, demonstrating that several isolate release glycerol albeit at levels substantially lower than those that induce differentiation. The rationale for the focus on fatty acids in the discussion and other requested clarifications has been met.

We thank the reviewer for the positive feedback.

Reviewer #3:

Remarks to the Author:

The authors have dealt appropriately with my comments on the originally submitted manuscript. The modifications made in response to my previous comments and those by the reviewers have improved the manuscript and strengthened the conclusions drawn.

I just spotted a few very minor issues:

1. Page 8 and Fig. 2: No reference to Figures 2B and 2C is made in the text.
2. Line 428: trypanocidal should be trypanocidal.
3. Line 512: Fig. 9H? This figure does not exist. Should it be 6F?
4. Line 801: Add 'glycerol' after (B).
5. Lines 820 and 824: The legend mentioned 'release of fatty acids', but Supplementary Figures 5A and 5D show release (or rather presence) of glycerol.
6. Line 834: 'fatty acids' should be replaced by 'glycerol' (shown in Supplementary Fig. 6).

We thank the reviewer for pointing these errors, which have been corrected.

Final Decision Letter:

Message 11th September 2023

:
Dear Luisa,

I am pleased to accept your Article "Adipocyte lipolysis protects mice against Trypanosoma

23brucei infection" for publication in Nature Microbiology. Thank you for having chosen to submit your work to us and many congratulations.

Acceptance of your manuscript is conditional on all authors' agreement with our publication policies (see <https://www.nature.com/nmicrobiol/editorial-policies>). In particular your manuscript must not be published elsewhere and there must be no announcement of the work to any media outlet until the publication date (the day on which it is uploaded onto our website).

Please note that *Nature Microbiology* is a Transformative Journal (TJ). Authors may publish their research with us through the traditional subscription access route or make their paper immediately open access through payment of an article-processing charge (APC). Authors will not be required to make a final decision about access to their article until it has been accepted. [Find out more about Transformative Journals](https://www.springernature.com/gp/open-research/transformative-journals)

Authors may need to take specific actions to achieve [compliance with funder and institutional open access mandates](https://www.springernature.com/gp/open-research/funding/policy-compliance-faqs). If your research is supported by a funder that requires immediate open access (e.g. according to [Plan S principles](https://www.springernature.com/gp/open-research/plan-s-compliance)) then you should select the gold OA route, and we will direct you to the compliant route where possible. For authors selecting the subscription publication route, the journal's standard licensing terms will need to be accepted, including [self-archiving policies](https://www.nature.com/nature-portfolio/editorial-policies/self-archiving-and-license-to-publish). Those licensing terms will supersede any other terms that the author or any third party may assert apply to any version of the manuscript.

With kind regards,